# *One for all and all for one:* EFFICIENT COMPUTATION OF PARTIAL WASSERSTEIN DISTANCES ON THE LINE

**Laetitia Chapel**
L'Institut Agro Rennes-Angers – IRISA
Rennes, France
`laetitia.chapel@irisa.fr`

**Romain Tavenard**
Université de Rennes 2 – IRISA
Rennes, France
`romain.tavenard@univ-rennes2.fr`

## ABSTRACT

Partial Wasserstein helps overcoming some of the limitations of Optimal Transport when the distributions at stake differ in mass, contain noise or outliers or exhibit mass mismatches across distribution modes. We introduce PAWL, a novel algorithm designed to efficiently compute exact PArtial Wasserstein distances on the Line. PAWL not only solves the partial transportation problem for a specified amount of mass to be transported, but *for all* admissible ones. This flexibility is valuable for machine learning tasks where the level of noise is uncertain and needs to be determined through, e.g., cross-validation. By achieving $\mathcal{O}(n \log n)$ time complexity for the partial 1-Wasserstein problem on the line, it enables practical applications with large scale datasets. Additionally, we introduce a novel slicing strategy tailored to Partial Wasserstein, which does not permit transporting mass between outliers or noisy data points. We demonstrate the advantages of PAWL in terms of computational efficiency and performance in downstream tasks, outperforming existing (sliced) Partial Optimal Transport techniques.

## 1 INTRODUCTION

Optimal Transport (OT) plays a crucial role in machine learning (ML) by offering geometry-aware ways to compare probability distributions. To name a few flagship applications, optimal transport and the Wasserstein distance have being used as a loss in classification (Frogner et al., 2015) and generative modeling (Arjovsky et al., 2017), can be used for quantization in unsupervised learning (Cuturi & Doucet, 2014), and allows to align distributions in domain adaptation (Courty et al., 2016). However, OT suffers from two major limitations that may prevent its practical use in ML. First, its sensitivity to noise in the input distributions or mass mismatches between their modes hinder a reliable computation of the distance and can degrade the performance of downstream tasks. Second, existing OT solvers struggle with respect to computational efficiency. Most algorithms exhibit polynomial time complexity, making OT unsuitable for large-scale applications.

Unbalanced Optimal Transport (UOT (Benamou, 2003), see Séjourné et al. (2023b) for a review) is often deemed to be more robust to noise or outliers than its balanced counterpart (Balaji et al., 2020). UOT relaxes the mass conservation constraint of OT, allowing to match subparts of distributions or distributions with unequal total masses. Partial Optimal Transport (Caffarelli & McCann, 2010) is a special case of UOT, on which the amount of mass to be transferred can be fixed, which is a convenient feature when the amount of out-of-distribution samples is known beforehand. Note also that Unbalanced or Partial OT can come with a *profile* (Phatak et al., 2023) or a *regularization path* (Chapel et al., 2021) that provides solutions for all values of the method's parameter at once. This can be leveraged for parameter selection or faster cross-validation on the mass to be transferred.

Exact solvers for OT (and UOT) exhibit cubic complexity in the number of samples, making them impractical for large-scale applications. Various relaxations and approximations have been introduced to reduce this computational burden. In this paper, we rely on the *sliced* approximation, which builds upon the property that the Wasserstein distance between 1-dimensional distributions can be computed in $\mathcal{O}(n \log n)$ (Peyré et al., 2019). For UOT on the line, several algorithms have been devised to efficiently find solutions in general cases (Bai et al., 2023; Séjourné et al., 2023a), when the transport is an injective map (Bonneel & Coeurjolly, 2019) or involves a tree metric (Sato et al., 2020).

In this paper, we aim to improve the usability of optimal transport by introducing a novel algorithm for efficiently computing PArtial Wasserstein distances on the Line (PAWL) along with a well-grounded, original slicing scheme tailored for partial OT. Unlike previous approaches with polynomial complexities, PAWL *exactly* solves the partial 1-Wasserstein problem in 1d in $\mathcal{O}(n \log n)$ time complexity when the ground cost is the Manhattan distance, making it suitable for practical use cases involving sliced partial OT on large-scale datasets. It provides solutions for all transported masses at once, empowering users to accommodate diverse tasks when hyper-parameter tuning is required. Before diving into the details of our PAWL solver, we review the necessary background on Optimal Transport. We then introduce the proposed algorithm and the tailored slicing scheme. Empirical evaluations are provided in Section 5.

## 2 BACKGROUND ON OPTIMAL TRANSPORT

Let us consider two point clouds $\{\boldsymbol{x}_i\}_{i=1}^n$ and $\{\boldsymbol{y}_j\}_{j=1}^m$ where each sample belongs to $\mathbb{R}^d$, and the associated empirical measures $\mu = \sum_i a_i \delta_{\boldsymbol{x}_i}$ and $\nu = \sum_j b_j \delta_{\boldsymbol{y}_j}$ with $a_i$ (resp. $b_j$) the mass associated to $\boldsymbol{x}_i$ (resp. $\boldsymbol{y}_j$). We assume we can define a ground cost $d(\boldsymbol{x}_i, \boldsymbol{y}_j)$ between $\boldsymbol{x}_i$ and $\boldsymbol{y}_j$. In Kantorovitch's formulation of the discrete optimal transport problem, one aims at transporting the total mass of the source $\mu$ to the target $\nu$ with a minimal cost:

$$\text{OT}(\mu, \nu) = \min_{\boldsymbol{\pi} \in \Pi(\mu,\nu)} \sum_{i,j} d(\boldsymbol{x}_i, \boldsymbol{y}_j)\pi_{ij} \qquad \text{(Kantorovitch)}$$

where the constraint set $\Pi(\mu, \nu) = \{\boldsymbol{\pi} \in \mathrm{R}_{n \times m}^+ \text{ such that } \forall i, \sum_j \pi_{ij} = a_i \text{ and } \forall j, \sum_i \pi_{ij} = b_j\}$. This problem admits a solution as long as $\mu$ and $\nu$ have the same total mass $\sum_i a_i = \sum_j b_j$. Matrix $\boldsymbol{\pi}$ is called the *transportation matrix* and $\pi_{ij}$ indicates how much mass is transported from $\boldsymbol{x}_i$ to $\boldsymbol{y}_j$. If $d(\boldsymbol{x}, \boldsymbol{y}) = \|\boldsymbol{x} - \boldsymbol{y}\|_p^p$, OT is a distance, called the $p$-Wasserstein distance.

Kantorovitch's formulation is a relaxation of the original Monge problem:

$$\min_{\mathbf{T}} \sum_i d(\boldsymbol{x}_i, \mathbf{T}(\boldsymbol{x}_i)) \qquad \text{(Monge)}$$

where $\mathbf{T} : \sum_i a_i \delta_{\mathbf{T}(\boldsymbol{x}_i)} = \sum_j b_j \delta_{\boldsymbol{y}_j}$ is called a *Monge map*. The question of the existence of such a map has been thoroughly studied: it requires, among other conditions, that $n \geq m$ and still that $\sum_i a_i = \sum_j b_j$. When $n = m$ and $a_i = b_j, \forall i, j$, it corresponds to the optimal assignment problem where one aims at finding a permutation operator $\sigma$ solving

$$\min_{\sigma} \frac{1}{n} \sum_i d(\boldsymbol{x}_i, \boldsymbol{y}_{\sigma(i)}). \qquad \text{(Assignment)}$$

### 2.1 PARTIAL OPTIMAL TRANSPORT

If the measures $\mu$ and $\nu$ are not normalized or are contaminated with outliers, one may instead solve an *unbalanced* optimal transport (UOT) problem, in which only a fraction $s \leq \min(\sum_i a_i, \sum_j b_j)$ of the mass is transported. Here, we focus on the *partial* Wasserstein (PW) problem, a particular instance of UOT, where the constraint set in Equation (Kantorovitch) is modified to $\Pi(\mu_{\leq}, \nu_{\leq}) = \{\boldsymbol{\pi} \in \mathrm{R}_{n \times m}^+ \text{ s. t. } \forall i, \sum_j \pi_{ij} \leq a_i, \forall j, \sum_i \pi_{ij} \leq b_j \text{ and } \sum_{ij} \pi_{ij} = s\}$. This problem was initially studied by Caffarelli & McCann (2010) who proved uniqueness of the solution for any cost function $d(\boldsymbol{x}, \boldsymbol{y}) = h(\boldsymbol{x} - \boldsymbol{y})$ (where $h$ is a convex function) in the case of measures with disjoint supports. Later, Figalli (2010) refined this result, relaxing the disjoint support assumption in the case of a quadratic cost function. He analyzed the function that associates to each mass $s$ the cost: $s \rightarrow \text{PW}(s)$.

Although the problem of deriving an UOT version of the Kantorovitch formulation has been well studied, deriving an analogous formulation for the Monge problem (Monge) is less obvious. Eyring et al. (2023) defined unbalanced Monge maps by rescaling the target measure. Gallouët et al. (2021) proved the existence of UOT maps in the context of specific cost function.

In this paper, we consider a variant of the linear assignment problem where all samples have equal mass, $a_i = b_j = w$, no constraints on $n$ and $m$, and only a fraction $s$ of the total mass is transported:

$$\text{PW}(\mu, \nu, s) = \min_{\boldsymbol{\pi} \in \mathrm{R}_{n \times m}^+} \sum_{i,j} d(\boldsymbol{x}_i, \boldsymbol{y}_j)\pi_{ij} \text{ such that } \forall i, j, \pi_{ij} \leq w \text{ and } \sum_{i,j} \pi_{ij} = s. \qquad \text{(PW(s))}$$

## 2.2 COMPUTATIONAL (PARTIAL) OPTIMAL TRANSPORT

Solving OT problems is computationally demanding, with exact solvers typically having a cubic complexity. One key for permitting a broader use of OT is to focus on favorable cases or approximate solutions that reduce this prohibitive complexity. For instance, the entropy-regularized formulation proposed by Cuturi (2013) reduces the complexity to $\mathcal{O}\left(n^2\right)$. When samples $x_i$ and $y_j$ are 1-dimensional, the complexity is $\mathcal{O}\left(n \log n\right)$ as a closed-form solution exists. It suffices to sort the distributions $\mu$ and $\nu$ and match their cumulative distribution functions. This special case is at the core of sliced OT (Rabin et al., 2012) which is defined as the average of 1 dimensional OT costs computed along $L$ projection directions over the unit-sphere. This approximation of OT has a complexity of $\mathcal{O}\left(dLn + Ln \log n\right)$, where $d$ is the dimension of the samples, and exhibits favorable statistical properties (e.g. improved sample complexity) that makes it a standard tool for the OT practitioner.

When it comes to partial OT, several algorithms have been devised. Phatak et al. (2023) propose an exact solver that computes partial OT solutions for all possible values of the mass $s$ with complexity $\mathcal{O}\left(n^3\right)$. Bai et al. (2023) define an efficient computation scheme for the partial OT problem on the line, with a quadratic worst-case complexity. Bonneel & Coeurjolly (2019) solve a specific one-dimensional injective partial assignment problem with a quasi linear time algorithm. Séjourné et al. (2022) introduce an iterative algorithm that converges to the exact solution of the 1d UOT problem with a quasi-linear complexity, but it does not extend to Partial OT, preventing its use when one aims at transporting a fixed amount of mass (rather than setting a hyper-parameter) or when a sparse transport plan is sought. In addition, plugging these 1d solvers into a slicing scheme is not obvious: as it corresponds to averages over different directions, different samples can be selected for each direction, which may lead to sub-optimal results (Séjourné et al., 2023a).

In this paper, we propose a new numerical scheme to compute partial solutions for *all* possible values of the transported mass $s$ with a $\mathcal{O}\left(n \log n\right)$ complexity for the Manhattan cost (i.e. we focus on the partial 1-Wasserstein problem). We also define an appropriate slicing scheme based on Mahey et al. (2024) that allows an efficient and grounded sliced partial OT solution.

## 3 PARTIAL WASSERSTEIN DISTANCES ON THE LINE (PAWL)

Let $\boldsymbol{x} = \{x_1, \cdots, x_n\}$ and $\boldsymbol{y} = \{y_1, \cdots, y_m\}$ be 1d empirical arbitrary measures, with fixed weights $a_i = b_j = w$. As in Caffarelli & McCann (2010), we assume that the two distributions are disjoint sets, which allows ensuring the existence and uniqueness of the solution. In the following, we denote by $\boldsymbol{z} = \{z_1, \cdots, z_{n+m}\}$ the union distribution and we assume that $\boldsymbol{x}, \boldsymbol{y}$ and $\boldsymbol{z}$ have been sorted in a preprocessing step, which can be done in $\mathcal{O}\left(n \log n\right)$.

The overall idea of PAWL is to be able to efficiently compute solutions $\boldsymbol{\pi} \in \Pi(\mu_{\leq}, \nu_{\leq})$ for *all* possible amounts of transported mass $s$ whenever the set of constraint is not empty. To do so, let us introduce the related problem for a given number $k$ of samples to be transported:

$$\min_{\boldsymbol{\pi}} \sum_{i,j} d(x_i, y_j)\pi_{ij} \text{ such that } \forall i, j, \pi_{ij} \in \{0, w\} \text{ and } \sum_{i,j} \pi_{ij} = k \cdot w. \qquad \text{(PAWL}_k\text{)}$$

We denote $\boldsymbol{\pi}^k$ the $\arg\min$ of PAWL$_k$.

**Proposition 1.** *To solve problem PW(s) on the line, denoted PWL(s), it is sufficient to solve problems PAWL$_k$ and PAWL$_{k'}$, with $k' = \lceil \frac{s}{w} \rceil$ and $k = \lfloor \frac{s}{w} \rfloor$ and $\boldsymbol{\pi}(s) = \boldsymbol{\pi}^k + (\boldsymbol{\pi}^{k'} - \boldsymbol{\pi}^k) \cdot (s \mod w)$.*

Proofs for all propositions are provided in Appendix. In the following, our goal will hence be to design an iterative algorithm that will construct $\{\boldsymbol{\pi}^k\}_k$ for growing values of $k$. We denote $\mathcal{A}_k = \left\{x_i : \sum_j \pi_{ij}^k > 0\right\} \cup \left\{y_j : \sum_i \pi_{ij}^k > 0\right\}$ the *active set* associated to a solution $\boldsymbol{\pi}^k$ for PAWL$_k$.

### 3.1 A FIRST, $\mathcal{O}\left(n^4\right)$, ALGORITHM

Our PArtial Wasserstein distances on the Line (PAWL) solver will rely on induction to compute, incrementally, solutions for PAWL$_k$ with growing values of $k$. The main ingredient for this construction is the following important result from Caffarelli & McCann (2010):

**Theorem 1.** *For all $k < k'$, there exist solutions for PAWL$_k$ and PAWL$_{k'}$ such that $\mathcal{A}_k \subset \mathcal{A}_{k'}$.*

**Initialization.** We aim to compute a solution for $\text{PAWL}_0$. The constraint set on $\boldsymbol{\pi}(0)$ is such that the only admissible solution is $\pi_{ij} = 0$ for any $i, j$. Hence $\text{PAWL}_0 = 0$ and $\mathcal{A}_0 = \emptyset$.

**Induction.** Let us now assume that a solution for $\text{PAWL}_k$ has been constructed. Our goal is now to solve $\text{PAWL}_{k+1}$. Using straightforward cardinality reasoning on active sets considered in Theorem 1, there exists $\mathcal{A}_{k+1}$ of the form $\mathcal{A}_k \cup \{x_i, y_j\}$, with $x_i, y_j \in \overline{\mathcal{A}_k}$ the complement of $\mathcal{A}_k$. A naive solution would then be to iterate over all candidate pairs and pick the one that least increases the cost.

In the following, we will aim at improving the induction step of this naive algorithm in two decisive ways. The first one is to drastically restrict the candidate pairs set and the second one to efficiently compute the increase in cost each candidate pair would induce. Section titles reflect the successive gains in time complexity offered by each improvement.

## 3.2 RESTRICTING THE CANDIDATE SET TO GET $\mathcal{O}\left(n^3\right)$ TIME COMPLEXITY

Let us first restrict the set of candidate pairs $\{x_i, y_j\}$ at a given step $k$ of our algorithm. To do so, we rely on the following property, illustrated in Figure 1:

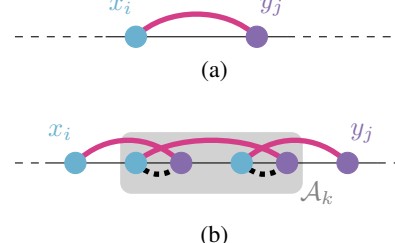

**Proposition 2.** *A given sample $z_\ell \in \overline{\mathcal{A}_k}$ can only be added in the active set together with one of its neighbors in $\overline{\mathcal{A}_k}$ coming from the other distribution.*

This property restrains the search space from $\mathcal{O}\left((n - k)^2\right)$ enumerations to $\mathcal{O}\left(\text{Card}(\overline{\mathcal{A}_k})\right) = \mathcal{O}(n - k)$. For each pair, computing the cost of the candidate active set $\mathcal{A}_k \cup \{x_i, y_j\}$ is a 1d OT problem on already sorted distributions, for which complexity is $\mathcal{O}(k)$. Induction step $k$ hence runs in $\mathcal{O}(k(n - k))$ time.

At this point, the algorithm iteratively constructs solutions for all $\text{PAWL}_k$ problems for $k$ ranging from 0 to $n$, with an overall time complexity of $\mathcal{O}\left(n^3\right)$. This complexity will be further lowered in Section 3.4 by providing ways

Figure 1: Candidate pairs for $\text{PAWL}_{k+1}$ given $\mathcal{A}_k$. At each iteration $k$, a candidate pair is either a pair of neighbors in the union distribution (a) or it is such that all points in between are already included in $\mathcal{A}_k$ (b). The dotted links are the matchings in $\text{PAWL}_k$ and those in magenta are the matchings in $\text{PAWL}_{k+1}$.

of accessing the best candidates $\{x_i, y_j\}$ at each step. For now, let us have a closer look at what it takes to efficiently compute the cost of including a given candidate pair.

## 3.3 EFFICIENTLY COMPUTING THE OT COST FOR EACH CANDIDATE SET ($\mathcal{O}\left(n^2\right)$)

For each step, we now need to compute the increase in cost induced by each pair in the (restricted) set of candidates. We provide an $\mathcal{O}(1)$ method to compute this increase in cost for any candidate when $d(\cdot, \cdot)$ is the Manhattan distance. In a nutshell, we will show that i) all candidates form specific patterns on the real line coined *chains* and that such chains can be extracted efficiently, ii) those chains can be decomposed into minimal sets for which the OT cost can be computed in $\mathcal{O}(1)$ and iii) it is possible to examine only one candidate at each iteration of our algorithm.

As stated earlier, candidate pairs are of the form $\{x_i, y_j\}$ where $x_i$ and $y_j$ are neighbors in $\overline{\mathcal{A}_k}$. This can only happen in two settings: (S1) $x_i$ and $y_j$ are also neighbors in $\boldsymbol{z}$ (i.e. there exists $\ell$ such that $\{x_i, y_j\} = \{z_{\ell-1}, z_\ell\}$) or (S2) $x_i$ and $y_j$ are not neighbors in $\boldsymbol{z}$ and all points from $\boldsymbol{z}$ included in the open interval $(x_i, y_j)$ are in $\mathcal{A}_k$ (i.e. $\{x_i, y_j\} = \{z_{\ell-p}, z_\ell\}$ and $z_{\ell-p+1}, \ldots, z_{\ell-1} \in \mathcal{A}_k$).

In order to efficiently compute the increment in cost induced by such candidate sets, we will rely on the notion of chain defined hereafter:

**Definition 1** (Chain $\mathscr{C}$). *A chain $\mathscr{C}$ is an ordered set of* contiguous *samples that is balanced, i.e. such that the total number of samples from $\mathbf{x}$ in $\mathscr{C}$ is equal to the total number of samples from $\mathbf{y}$ in $\mathscr{C}$. This notion is illustrated in magenta in Figure 2.*

**Definition 2** (Cost of a chain $c(\mathscr{C})$). *Let $\mathscr{C}$ be a chain. Its cost is $c(\mathscr{C}) = OT(\mu_\mathscr{C}, \nu_\mathscr{C})$ where $\mu_\mathscr{C} = \sum_i \mathbb{1}_{(\boldsymbol{x}_i \in \mathscr{C})} a_i \delta_{\boldsymbol{x}_i}$ and $\nu_\mathscr{C} = \sum_j \mathbb{1}_{(\boldsymbol{y}_j \in \mathscr{C})} b_j \delta_{\boldsymbol{y}_j}$.*

Let us first focus on the (S1) setting. As shown in the proof for Proposition 2, $\mathcal{A}_k$ is the union of disjoint chains. As a consequence, knowing that samples are mapped in order, $x_i$ will be mapped to

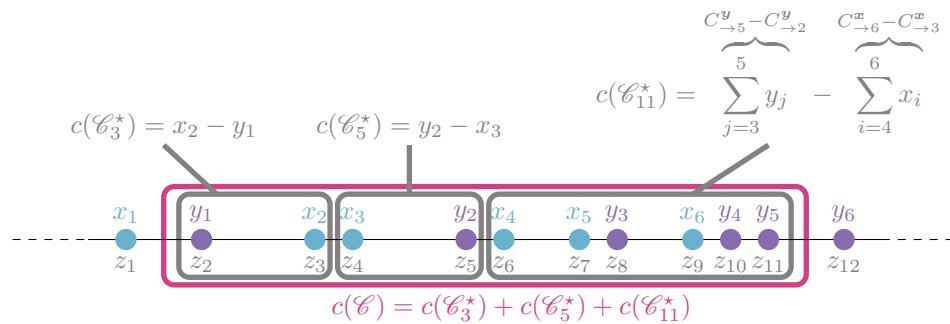

Figure 2: Chains and their associated costs. In order to compute the cost for chain $\mathscr{C}$ (in magenta), we partition it into a sequence of 3 minimal chains $\mathscr{C}_3^\star, \mathscr{C}_5^\star$ and $\mathscr{C}_{11}^\star$, for which cost computation can be made efficient when the Manhattan distance is considered.

$y_j$ in $\mathcal{A}_{k+1}$ and the increment in the transportation cost between $\mathcal{A}_k$ and $\mathcal{A}_k \cup \{x_i, y_j\}$ will be the distance between $x_i$ and $y_j$ (cf Figure 1a). All such distances can be computed in a preprocessing step with complexity $\mathcal{O}(n)$ such that no additional cost is incurred at step $k$.

When it comes to the (S2) setting, the increase in cost cannot be computed the same way, since some of the mappings involved in $\text{PAWL}_k$ will be revoked in $\text{PAWL}_{k+1}$. More precisely, all the mappings involving samples in the interval $[x_i, y_j]$ will be revoked (cf Figure 1b). We now show how to efficiently compute the increment in cost induced by changing the active set from $\mathcal{A}_k$ to $\mathcal{A}_k \cup \{x_i, y_j\}$. The set $\{z_{\ell-p}, \ldots, z_\ell\}$ introduced in (S2) is a balanced set of contiguous samples. It is hence a chain, denoted $\mathscr{C}_{\ell-p \to \ell}$. Note that $\mathscr{C}_{\ell-p+1 \to \ell-1}$ is also a chain. In what follows, the increment in cost between $\mathcal{A}_k$ and $\mathcal{A}_k \cup \{x_i, y_j\}$ will be referred to as the *marginal cost* associated to $\mathscr{C}_{\ell-p \to \ell}$. It can be computed as

$$\text{marginal\_cost}(\mathscr{C}_{\ell-p \to \ell}) = c(\mathscr{C}_{\ell-p \to \ell}) - c(\mathscr{C}_{\ell-p+1 \to \ell-1}). \tag{1}$$

**Efficiently computing the cost of a chain.** As can be seen in Equation (1), being able to efficiently compute the cost of any chain is sufficient to ensure efficient computation of the marginal cost of any chain. In order to allow $\mathcal{O}(1)$ computation of the cost of chains during the run of our algorithm, we will precompute sufficient quantities such that the cost of a chain can be accessed through simple lookups, relying the following proposition:

**Proposition 3.** *Let $\mathscr{C}_{\ell-p \to \ell}$ be a chain. Then we have*

$$c(\mathscr{C}_{\ell-p \to \ell}) = c(\mathscr{C}_{\cdot \to \ell}) - c(\mathscr{C}_{\cdot \to \ell-p-1}).$$

*Notation $\mathscr{C}_{\cdot \to \ell}$ denotes the* maximal *chain with a largest element at position $\ell$, i.e. $\mathscr{C}_{\cdot \to \ell} = \arg\min_{\mathscr{C}_{i \to \ell}} \text{card}(\mathscr{C}_{i \to \ell})$ and we use the convention $c(\mathscr{C}_{\cdot \to \ell-p-1}) = 0$ if there is no chain with a largest element at position $\ell - p - 1$.*

The burden of precomputing the cost of any chain has now shifted to that of computing, for any position $\ell$, the cost of the maximal chain with largest position $\ell$. We now derive an algorithm for the computation of maximal chain cost, based on the observation that a chain is a set of contiguous non overlapping minimal chains, where minimal chains are defined as:

**Definition 3** (Minimal chain $\mathscr{C}_\ell^\star$). *A chain $\mathscr{C}$ with its largest element at position $\ell$ is said to be minimal iff it is the* smallest *set of* contiguous *points with a largest element at position $\ell$ in $\mathbf{z}$ that is balanced. For each position $\ell$, there exists* at most *one minimal chain, that we denote $\mathscr{C}_\ell^\star$.*

This notion of minimal chain is illustrated in gray in Figure 2. We have:

$$c(\mathscr{C}_{\cdot \to \ell}) = \begin{cases} c(\mathscr{C}_\ell^\star) + c(\mathscr{C}_{\cdot \to \ell'-1}) & \text{if a minimal chain of the form } \mathscr{C}_\ell^\star = \{z_{\ell'}, \ldots, z_\ell\} \text{ exists} \\ 0 & \text{otherwise.} \end{cases} \tag{2}$$

Given this, and assuming that we can access the existence of a minimal chain with a maximal element at any position, one can compute all $c(\mathscr{C}_{\cdot \to \ell})$ terms in increasing order of $\ell$ using dynamic programming in $\mathcal{O}(n)$. Once these quantities stored, the cost for any chain can be computed in $\mathcal{O}(1)$ using Proposition 3. We now need to extract all minimal chains and their associated cost from $\mathbf{z}$.

---

**Algorithm 1:** Efficient computation of chain costs for the Manhattan distance

---

**Data:** Sorted $x$, Sorted $y$, Sorted $z$

1 Compute all cumulative sums of the form $C^{\mathbf{x}}_{\to i}$ and $C^{\mathbf{y}}_{\to j}$ ▷ `Precomputations`

2 Extract all minimal chains from $z$ ▷ `Uses Proposition 4`

3 **for** $j \in [1..n]$ **do**

4  $\quad$ **if** $\mathscr{C}^{\star}_j$ *exists* **then**

5  $\quad\quad$ | Compute $c\left(\mathscr{C}^{\star}_j\right)$ using Proposition 5 and the necessary cumulative sums

6  $\quad$ **end**

7  $\quad$ Compute $c\left(\mathscr{C}_{\cdot \to j}\right)$ using Equation (2)

8 **end**

$\quad$ ▷ `Computation of a chain marginal cost using eq. 1 and prop. 3`

9 $\text{marginal\_cost}\left(\mathscr{C}_{\ell-p \to \ell}\right) \leftarrow c\left(\mathscr{C}_{\cdot \to \ell+1}\right) - c\left(\mathscr{C}_{\cdot \to \ell-p-1}\right) - c\left(\mathscr{C}_{\cdot \to \ell}\right) + c\left(\mathscr{C}_{\cdot \to \ell-p}\right)$

---

**Efficiently computing all minimal chains and their associated costs.**

**Proposition 4.** *Assuming that $z$ is sorted, extracting all minimal chains can be done in $\mathcal{O}\left(n\right)$ by iterating over items in $z$ in increasing order.*

We also need to efficiently compute minimal chain costs. If an arbitrary distance function is used, then computation of the cost for a chain of length $L$ can be performed in $\mathcal{O}\left(L\right)$ by summing distances of items mapped in increasing order. Let us now focus on the specific case of the Manhattan distance. First, one should notice that, inside a minimal chain, all mappings have the same ordering (cf. Proposition 5's proof). In other words, if $\mathscr{C}^{\star}$ is a minimal chain and $\{x_i, y_j\}$ is one of the mappings inside $\mathscr{C}^{\star}$ with $x_i < y_j$, then all the mappings in $\mathscr{C}^{\star}$ will be of the form $\{x, y\}$ with $x < y$. The following proposition holds as a direct consequence of this observation:

**Proposition 5.** *Let us denote the cumulative sums $C^{\mathbf{x}}_{\to \ell} = \sum_{i \leq \ell} z_i \mathbb{1}_{z_i \in \mathbf{x}}$ and similarly $C^{\mathbf{y}}_{\to \ell} = \sum_{i \leq \ell} z_i \mathbb{1}_{z_i \in \mathbf{y}}$, with $C^{\mathbf{x}}_{\to 0} = C^{\mathbf{y}}_{\to 0} = 0$ by convention. The Manhattan cost for the minimal chain $\mathscr{C}^{\star}_{\ell} = \{z_{\ell-p}, \ldots, z_{\ell}\}$ is:*

$$c\left(\mathscr{C}^{\star}_{\ell}\right) = \left| C^{\mathbf{x}}_{\to \ell} - C^{\mathbf{y}}_{\to \ell} - C^{\mathbf{x}}_{\to \ell-p-1} + C^{\mathbf{y}}_{\to \ell-p-1} \right|$$

*which can be computed in $\mathcal{O}\left(1\right)$ time once the cumulative sums stored.*

At that point, we can store, as a preprocessing step, all minimal chains and their associated costs in a lookup table such that one can access the cost and smallest element of a minimal chain of largest position $\ell$ in $\mathcal{O}\left(1\right)$ (lines 1–6 in Alg. 1). We have reduced the complexity of induction step $k$ from $\mathcal{O}\left(k(n-k)\right)$ to $\mathcal{O}\left(n-k\right)$, which leads to an improved complexity of $\mathcal{O}\left(n^2\right)$ to compute solutions to all partial problems. Our next step will be to avoid iterating through all candidates at each step.

## 3.4 CHOOSING THE BEST CANDIDATE AT EACH STEP ($\mathcal{O}\left(n \log n\right)$)

At each step $k$, we have to decide which pair is the best candidate. In order not to iterate over all candidates at each step, we maintain an ordered list of candidates, sorted in increasing order of their marginal cost. Choosing the best pair at each step then reduces to popping the first element of the list.

Recall that the candidates are either (S1) pairs of neighbors in $z$ or (S2) such that all intermediate points are already in $\mathcal{A}_k$ (Figure 1). We will hence make sure that all such pairs are included in our sorted list of candidates before they can be involved in an active set. First, we initialize the list with all neighbor pairs, for which distances can be computed in $\mathcal{O}\left(n\right)$. For these candidates, the marginal cost is their pairwise distance. Second, after each step $k$ of the algorithm, assuming that the pair $\{x_i, y_j\}$ has been selected as the best candidate, we can extract the largest chain $\mathscr{C}$ such that $x_i, y_j \in \mathscr{C}$ and $\mathscr{C} \subseteq \mathcal{A}_k$. We call this chain the *covering chain* for $x_i, y_j$ in $\mathcal{A}_k$. If the immediate predecessor and

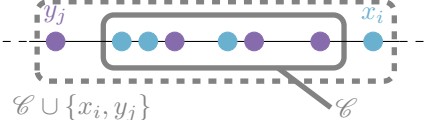

Figure 3: New candidate chain to be considered after step $k$. At step $k$, the chain $\mathscr{C}$ (solid line) has been formed in $\mathcal{A}_k$. $\mathscr{C} \cup \{x_i, y_j\}$ (dashed line) hence arises as a new candidate for future steps.

---

**Algorithm 2:** Efficient computation of all PArtial Wasserstein distances on the Line (PAWL)

---

**Data:** Sorted $\boldsymbol{z}, s$ (maximum mass to be transported)
**Result:** $\{\mathcal{A}_k\}_k$ and their associated costs

1  list_candidates $\leftarrow$ initial_candidates($\boldsymbol{z}$)    ▷ Sorted in ascending order of cost
2  $\mathcal{A}_0 \leftarrow \emptyset$
3  **for** $k \in \{1, 2, \ldots, \lceil \frac{s}{w} \rceil\}$ **do**
4     |   $x_i, y_j, c \leftarrow$ list_candidates.pop()   ▷ Candidate with minimal marginal cost
5     |   $\mathcal{A}_k \leftarrow \mathcal{A}_{k-1} \cup \{x_i, y_j\}$                         ▷ cost($\mathcal{A}_k$) = cost($\mathcal{A}_{k-1}$) + c
6     |   $l, m \leftarrow$ covering_chain_indices($x_i, y_j, \mathcal{A}_k$)
7     |   list_candidates.clean($\mathscr{C}_{l \to m}$) ▷ Remove candidates overlapping with $\mathscr{C}_{l \to m}$
8     |   **if** $z_{l-1}$ and $z_{m+1}$ *do not come from the same distribution* **then**
9     |      |   marginal_cost $\leftarrow c(\mathscr{C}_{l-1 \to m+1}) - c(\mathscr{C}_{l \to m})$
10    |      |   list_candidates.insert_sorted($z_{l-1}, z_{m+1}$, marginal_cost)
11    |   **end**
12  **end**

---

successor elements of $\mathscr{C}$ do not come from the same distribution, then the union of $\mathscr{C}$ and those two elements is a valid candidate for future steps $k' > k$, as shown in Figure 3: we can compute its induced increment in cost (cf. Algorithm 1) and insert it in the ordered list.

By doing so, at each step of our algorithm, we pop exactly one candidate from our list of candidates, and append at most one new candidate to that list. Since the list is initialized with at most $n$ elements, the whole process can be done in $\mathcal{O}(n \log n)$ complexity and this will dominate the complexity for Algorithm 2 when the Manhattan distance is used as a cost. For other costs, and assuming a single cost computation takes $\mathcal{O}(C)$ time, the overall cost is $\mathcal{O}(Cn^2)$ (see Appendix A.2 for more details). Finally, note that this algorithm provides active sets and transportation costs. Recovering the transportation plan for a given $k$ requires an additional $\mathcal{O}(k \log k)$ step to sort $\mathcal{A}_k$.

## 4   SLICED PARTIAL WASSERSTEIN

Sliced OT Rabin et al. (2012) leverages 1d solvers to approximate the Wasserstein distance between probability distributions. While various methods have been introduced for sliced partial Wasserstein distances in $\mathbb{R}^d$, there is currently no widely accepted consensus on the best approach within the community. Bai et al. (2023) define the Sliced Optimal Partial Transport that averages over 1d partial Wasserstein computed on the line, taking into account outliers into the approximation (note that, in their experiments, Bai et al. (2023) only rely on 1 single projection in order to avoid this side effect). Séjourné et al. (2023a) propose to reweight *globally* the input measures into the slicing process rather than projection by projection, allowing one to consistently discard some samples. However, their algorithm relies on specific properties of the UOT problem that prevent its extension to partial Wasserstein. Other variants of Sliced Wasserstein have been defined that either focus on refining the averaging process (Nguyen & Ho, 2024; Nguyen et al., 2024) or build upon the maximum distance over sampled projection directions (Deshpande et al., 2019). Here, we advocate to rely on Mahey et al. (2024) to define a sliced-PW that has the advantage to provide a sparse approximated transport map, hence allowing to identify samples as OOD. Our sliced-PW is a bi-level optimization problem, where the inner problem is a PWL($s$) one:

$$\text{Sliced-PW}(\mu, \nu, s) = \min_{\theta \in \mathbb{S}^d} \sum_{i,j} d(\boldsymbol{x}_i, \boldsymbol{y}_j) \pi_{ij}^{\star}(\theta) \text{ with } \boldsymbol{\pi}^{\star}(\theta) = \operatorname*{arg\,min}_{\boldsymbol{\pi} \in \Pi(\mu_{\leq}, \nu_{\leq})} \sum_{i,j} d(\langle \boldsymbol{x}_i, \theta \rangle, \langle \boldsymbol{y}_j, \theta \rangle) \pi_{ij}$$

such that $\pi_{ij} \leq w$ and $\sum \pi_{ij} = s$. $\langle \boldsymbol{x}_i, \theta \rangle$ and $\langle \boldsymbol{y}_j, \theta \rangle$ are the projections of the samples on to the direction $\theta$ over the unit sphere $\mathbb{S}^d$. In practice, we sample several directions and keep the one that gives the lowest PWL($s$) solution. Sliced-PW inherits the main properties of SWGG, that is to say:

**Proposition 6.** *Sliced-PW($\mu, \nu, s$) is an upper bound of PW and it is a semi-metric almost surely.*

## 5 EXPERIMENTAL VALIDATION

To evaluate the efficiency and practicality of PAWL[1], we conducted comprehensive experiments comparing it with the main state-of-the-art related solvers. We compare them in terms of computation times and usefulness in three different settings. PAWL computes solutions for all partial optimal transport problems at the same cost of computing only one. This advantage allows it to operate without any prior assumption about the amount of mass to be transported, enabling a post-hoc selection of this quantity through the elbow method based on the variation of the transportation cost with respect to the number of transported samples. In the following experiments, we compare PAWL with a fixed amount of mass to be transported with PAWL (elbow) that computes all partial optimal transports and decides on the amount of mass to be transported using an off-the-shelf implementation of the elbow method (Satopaa et al., 2011). As a result, this version does not require prior knowledge about the actual amount of corrupted or noisy samples, compared to other solvers.

### 5.1 COMPUTATIONAL EFFICIENCY

In order to validate the theoretical complexity of PAWL, we compare its running times with those obtained for OPT (Bai et al., 2023) that solves the exact same problem, and Fast-UOT (Séjourné et al., 2022), that solves a different yet related problem. In more details, OPT is an exact solver for POT given a hyper-parameter $\lambda$ that drives the amount of transported mass. Fast-UOT is an exact solver for the Kullback-

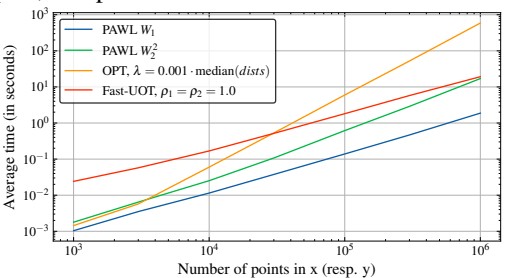

Figure 4: Running time comparisons for 1d partial optimal transport problems.

Leibler UOT problem and has one important hyper-parameter $\rho$ which controls the strength of the KL relaxation in the unbalanced formulation. In this set of experiments, we set PAWL's target transported mass $s$ from Algorithm 2 to 1 such that, for PAWL, *all* partial optimal transport problem solutions are computed, whereas only one solution (corresponding to hyper-parameters $\lambda$ and $\rho$) is computed for the baselines. We study the computation time as a function of the number of points in the 1d distributions involved. To do so, we generate random discrete distributions in 1d and vary the size of their support between $10^3$ and $10^6$. As advised in Bai et al. (2023), the OPT's $\lambda$ is set as a fraction of the median distance between samples. A fixed value is used for Fast-UOT's $\rho$ hyper-parameter. Figure 4 shows a quasi-linear trend for both PAWL with Manhattan cost and Fast-UOT, whereas OPT exhibits a quadratic trend for larger problem sizes, which is coherent with theoretical complexities. Note that PAWL with squared Euclidean cost is also competitive, even if it exhibits a super-linear trend. Fast-UOT is slightly more computationally expensive than PAWL in practice.

### 5.2 PARTIAL GRADIENT FLOWS

Starting from an initial distribution $\mu$, Wasserstein Gradient flows aim at driving it toward a target distribution $\nu$ by minimizing a displacement (or flow) over time. At each iteration, we minimize the loss $\mu(t) \mapsto OT(\mu(t), \nu)$. We consider two variants of OT losses: Sliced Wasserstein (SW) Rabin et al. (2012) and SW Generalized Geodesic (SWGG) Mahey et al. (2024), considering 100 projection directions. We display the results in Figure 5 by considering a transport with all samples (Vanilla OT) and only a partial transport using PAWL, in which either we set the amount of mass to be moved or select it using the elbow method. We run the experiment for 500 iterations, with a fixed learning rate of $5e^{-2}$. We consider a scenario of bimodal distributions with unbalanced modes, which is a typical failure case of vanilla OT. As expected, in this case, all the samples are transported, regardless of the cluster unbalance. SW with PAWL fails at providing a consistent flow as all the samples can be transported depending on the direction, leading to a non-sparse transport. On the opposite, Sliced-PW defined in Sec. 4 produces a consistent flow, where some samples are not transported at all (black symbols), even in the case where the amount of mass to be transported is determined thanks to the elbow. Note here that the gradient flow scheme converges to the target $\nu$ even if Sliced-PW is not a metric. Indeed, thanks to the partial scheme (in opposite to unbalanced OT for instance), the set of samples that belong to the solution is sparse, hence the set of samples $\mu(t)$ transported at each iteration remains stable, and Sliced-PW then benefits from the metric properties of vanilla SWGG.

---

[1]Code is available at https://github.com/rtavenar/partial_ot_1d

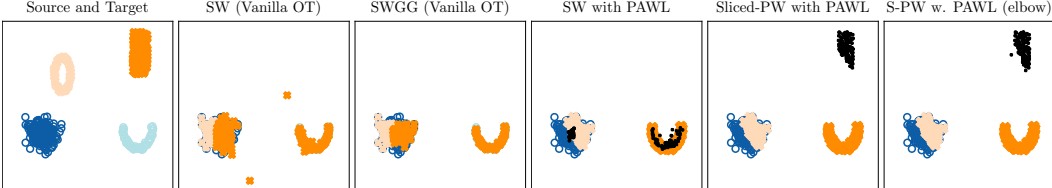

| Source and Target | SW (Vanilla OT) | SWGG (Vanilla OT) | SW with PAWL | Sliced-PW with PAWL | S-PW w. PAWL (elbow) |

Figure 5: Results of the Gradient Flows scheme for different sliced methods after 500 iterations. Orange symbols represents the source distribution and blue ones the target distribution. Light and dark symbol have different number of samples. In black, the samples that have not been transported at the last iteration. Sliced-PW is defined in Section 4.

Table 1: Performance for the point cloud registration experiment (error norms, the lower the better).

|  | Uses $s$? | 10k,5% | | | 10k,7% | | | 9k,5% | | | 9k,7% | | |
|---|---|---|---|---|---|---|---|---|---|---|---|---|---|
|  |  | 30s. | 60s. | End | 30s. | 60s. | End | 30s. | 60s. | End | 30s. | 60s. | End |
| SPOT | ✗ | **1.919** | **1.663** | 1.505 | **1.994** | 1.784 | 1.626 | **2.019** | **1.736** | 1.519 | **2.101** | **1.852** | 1.627 |
| PAWL (elbow) | ✗ | 2.155 | 1.754 | **0.014** | 2.097 | **1.621** | **0.030** | 2.359 | 2.104 | **0.121** | 2.340 | 2.097 | **0.174** |
| SOPT | ✓ | 1.242 | 0.451 | 0.001 | 1.066 | 0.426 | 0.002 | 1.090 | 0.566 | 0.167 | 1.120 | **0.621** | **0.212** |
| PAWL | ✓ | **0.301** | **0.207** | **0.000** | **0.511** | **0.356** | **0.000** | **0.689** | **0.510** | **0.116** | **0.992** | 0.702 | 0.318 |

## 5.3 POINT CLOUD REGISTRATION

Point cloud registration involves estimating the transformation between two sets of 3D points, a crucial task in various computer vision scenarios (Huang et al., 2021; Bai et al., 2023). Specifically, given two point clouds, we hypothesize an unknown mapping $T$ that relates them. In many cases, this transformation is constrained to follow the form $T(x) = \alpha Rx + \beta$, where $R$ is a rotation matrix, $\alpha > 0$ represents scaling, and $\beta$ is a translation vector. The objective is then to estimate $T$.

The standard method for this task is Iterative Closest Point (ICP Chen & Medioni (1992)) that alternates between matching points to their closest transformed neighbours and estimating the transform based on these matchings, until convergence. Bonneel & Coeurjolly (2019) introduces a variant of ICP in which the 1-nearest neighbour assignment is replaced by the sliced version of their 1d injective matching method, named SPOT. In the context of noisy point clouds, (Bai et al., 2023, Algorithm 3) introduce a variant relying on partial optimal transport to match distributions projected in 1d. Notably, parameters of the transform $T$ are estimated along the projection direction on transported samples alone. At each step of their algorithm, their hyper-parameter $\lambda$ is tuned such that the amount of transported samples is as close as possible to the actual mass of uncorrupted samples $s$, which is supposed to be known. For a fair comparison, we keep the same slicing scheme as it relies on one direction only and does not mixes several directions.

We will build on this strategy to develop two variants that rely on PAWL. A first variant, denoted "PAWL" is a direct adaptation of SOPT in which PAWL solver is plugged in place of the OPT one. Note that, in our case, we do not need to tune $\lambda$ at each step since we can use the prior about $s$ directly as an input. A second implementation, named "PAWL (elbow)" decides on the best amount of mass to be transported at each iteration.

We use the same noisy point cloud datasets as in Bai et al. (2023): Stanford Bunny, Dragon, Witch-Castle, Mumble Sitting. For each dataset, transforms are generated with uniformly sampled angles, translations, and scalings. Noisy points are appended to the distributions. Target data has a fixed size of 10k points; source data sizes vary (9-10k). The amount of added noisy samples ranges from 5% to 7%. We compare our PAWL variants to both SOPT (Bai et al., 2023) and SPOT (Bonneel & Coeurjolly, 2019). We do not include ICP in our comparison since it has been shown to perform poorly on noisy point clouds (Bai et al., 2023; Bonneel & Coeurjolly, 2019).

As shown in Table 1, both PAWL variants demonstrate low error in estimating the transform's parameters. When employing the prior on $s$, PAWL outperforms SOPT in rapid convergence and, at convergence, surpasses SOPT in 3 out of 4 cases. Notably, even without using prior information,

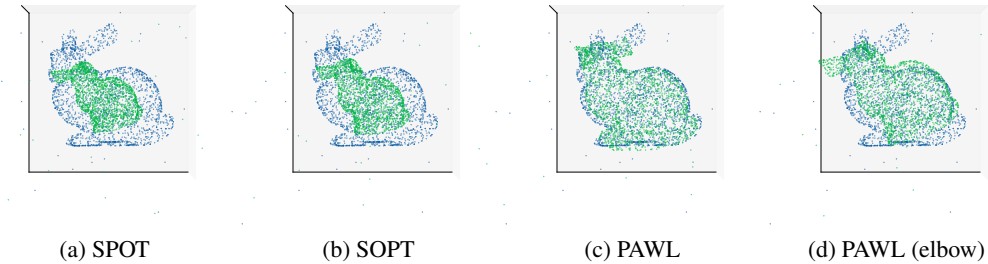

| (a) SPOT | (b) SOPT | (c) PAWL | (d) PAWL (elbow) |

Figure 6: Point cloud registration using SPOT, SOPT and our PAWL solver after 30 seconds. The target point cloud is in blue and the source point cloud is in green. The dataset used in this illustration is Stanford Bunny, with 10k samples in the source distribution and 7% of noisy points.

PAWL (elbow) is a solid competitor, exceeding SOPT (which relies on such prior information) in half the cases. However, PAWL (elbow) tends to require more time to converge due to the elbow method's initial challenge in accurately estimating the amount of samples to be transported when the transform is weakly estimated and clean/noisy samples are difficult to distinguish. These observations are corroborated by visual inspection of Figure 6.

## 5.4 SLICED PARTIAL WASSERSTEIN FOR DOMAIN ADAPTATION

We demonstrate the relevance of our Sliced-PW method, as defined in Section 4, in a higher-dimensional scenario. We consider the setup described in Chapel et al. (2021), where obtaining solutions for different regularization values proves useful for detecting data that may be contaminated with outliers. The source data consists of MNIST digits sampled from classes 0, 1, 2, 3 (200 points per class). The target data includes MNIST digits from classes 0 and 1, as well as Fashion MNIST digits from classes 8 and 9. Label propagation is used to classify the target data, for each possible mass amount $s$. Figure

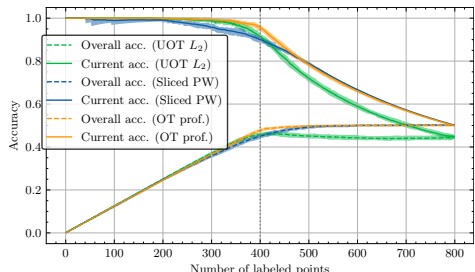

Figure 7: Evolution of the accuracies w.r.t. the number of transported samples.

7 compares the overall accuracy (defined as the ratio of correctly classified samples to the total number of samples) and the current accuracy (defined as the proportion of correctly classified samples among the transported points) of the Sliced-PW method and the regularization path approach proposed in Chapel et al. (2021) and the OT profiles (Phatak et al., 2023) (we considerered a $L_1$ ground cost), which are the state-of-the art methods that provide the entire set of solutions. Sliced-PW relies on $d + 100$ randomly drawn directions and for each $s$, it selects the direction of minimal cost. Experiments are repeated 10 times, and the results show that the average runtime of PAWL (1 sec. on a personal computer) is more than 200 times faster than that of UOT, and 14 times than OT profiles. Moreover, it provides comparable or even superior detection performance than UOT, as it avoids the need to set a threshold parameter to determine whether a sample has been transported.

## 6 CONCLUSION AND PERSPECTIVES

In this work, we have introduced PAWL, a novel algorithm for efficiently computing partial optimal transports between one-dimensional distributions. It enhances robustness against distributional variations (as Partial Wasserstein) while enabling practical use cases with large-scale datasets (as Wasserstein on the line). By computing solutions to all partial problems with a linear complexity, it allows an inspection of the evolution of the Wasserstein distance with the mass, hence providing a way to determine the best amount of mass to transfer.

Future work will explore differentiable variants of Sliced-PW: as we rely on the Manahattan distance to provide an efficient scheme, the smoothing scheme described in Mahey et al. (2024) can not be used, preventing the efficient search of an optimal slicing direction in large dimensions. Another line of work concerns the extension of PAWL to non constant masses $a_i \neq a_{i'}$ and $a_i \neq b_j$, or, ultimately, to continuous distributions. Finally, studying statistical guarantees on the subset that is selected through PW would allow establishing some conditions of convergence for our sliced-PW scheme.

ACKNOWLEDGMENTS

Both authors are deeply grateful to Paul Viallard for his invaluable feedback on early versions of this work. Romain Tavenard would like to acknowledge funding from ANR through project MATS ANR-18-CE23-0006.

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

# A  APPENDIX

## A.1  PROOFS

**Proposition 1.** *To solve problem PW(s) on the line, denoted PWL(s), it is sufficient to solve problems PAWL$_k$ and PAWL$_{k'}$, with $k' = \lceil \frac{s}{w} \rceil$ and $k = \lfloor \frac{s}{w} \rfloor$ and $\boldsymbol{\pi}(s) = \boldsymbol{\pi}^k + (\boldsymbol{\pi}^{k'} - \boldsymbol{\pi}^k) \cdot (s \mod w)$.*

*Proof of 1.* If $k = k'$, the proof is straightforward as $s = k \cdot w$. We then only focus on the case when $k \neq k'$.

Let $s$ be the mass to be transported, such that there exists $k$ with $k < \frac{s}{w} < k + 1$. From (Caffarelli & McCann, 2010, Prop 3.1), we know that there exist solutions to PAWL$_k$ and PAWL$_{k+1}$ such that their active sets are related by $\mathcal{A}_{k+1} = \mathcal{A}_k \cup \{x_i, y_j\}$. (Caffarelli & McCann, 2010, Theorem 3.4) (monotone expansion of active regions) tells us that there exists a solution $\boldsymbol{\pi}(s)$ to the problem PWL(s) such that the marginals of $\boldsymbol{\pi}$ are stable between $\boldsymbol{\pi}^k$, $\boldsymbol{\pi}(s)$ and $\boldsymbol{\pi}^{k+1}$, except for $\sum_\ell \pi_{i\ell}$ and $\sum_\ell \pi_{\ell j}$ (i.e. $x_i$ and $y_j$ are the only samples for which transported mass changes and for all other indices, the marginal is either 0 or $w$).

At this point, we hence know the marginals of a solution $\boldsymbol{\pi}(s)$. At fixed marginals, our Partial Wasserstein problem becomes a Wasserstein problem in 1d. It is known that these problems can be solved by matching cumulative distribution functions (Peyré et al., 2019, Equation 2.37), which can be obtained from the marginals of $\boldsymbol{\pi}$.

Let us assume, without loss of generality, that $x_i < y_j$. Since the marginals of $\boldsymbol{\pi}^k$, $\boldsymbol{\pi}(s)$ and $\boldsymbol{\pi}^{k+1}$ only differ in $x_i$ and $y_j$, they typically coincide outside the interval $[x_i, y_j]$. As a direct consequence, we have

$$\boldsymbol{\pi}(s) = \boldsymbol{\pi}^k + (\boldsymbol{\pi}^{k+1} - \boldsymbol{\pi}^k) \cdot (s \mod w)$$

outside this interval.

As discussed in greater details in Section 3.2, $x_i$ will be mapped with a sample inside the open interval $(x_i, y_j)$. More precisely, a direct application of matching the cumulative distribution functions gives that the mass $s \mod w$ coming from $x_i$ will be transported to the lowest $y$ sample that is greater than $x_i$. This sample $y$ used to be mapped to the lowest $x$ in $(x_i, y_j)$ in $\boldsymbol{\pi}^k$. Since it has only $w - (s \mod w)$ available mass remaining, it will send this mass to $x$. Since $x_i$ is mapped to $y$ in PAWL$_{k+1}$, the equality

$$\boldsymbol{\pi}(s) = \boldsymbol{\pi}^k + (\boldsymbol{\pi}^{k+1} - \boldsymbol{\pi}^k) \cdot (s \mod w)$$

holds for samples $x_i$ and $y$.

The exact same line of reasoning can be used recursively on the interval $(x, y_j)$ from which we exclude $y$ (since all of its mass has already been transported), *etc.* until we reach $y_j$. At this point, we have shown that the equality holds globally.  □

**Theorem 1.** *For all $k < k'$, there exist solutions for PAWL$_k$ and PAWL$_{k'}$ such that $\mathcal{A}_k \subset \mathcal{A}_{k'}$.*

*Proof of Theorem 1.* See (Caffarelli & McCann, 2010, Prop. 3.1) For two disjoint sets with finite cardinality and a continuous cost $d$, the active sets $\mathcal{A}_k$ grow monotonically with the amount of mass transferred $s = k \cdot w$.  □

**Proposition 2.** *A given sample $z_\ell \in \overline{\mathcal{A}_k}$ can only be added in the active set together with one of its neighbors in $\overline{\mathcal{A}_k}$ coming from the other distribution.*

*Proof of 2.* For a start, one should note that, if $\mathcal{A}_k$ is the active set for the PAWL$_k$ problem, then the mappings involved in the solution are those induced by the OT problem on distributions supported by $\boldsymbol{x} \cap \mathcal{A}_k$ and $\boldsymbol{y} \cap \mathcal{A}_k$ respectively, with uniform weights. Since we are in a 1-dimensional setting, we know that these couplings correspond to mapping the samples from both distributions in order.

We will now prove the following lemma:

**Lemma 1.** *If $\mathcal{A}_k$ is the active set for some POT$_k$ problem, then it can be written as the union of disjoint chains:*

$$\mathcal{A}_k = \bigcup_i \mathcal{C}_i \ \text{such that } \forall i \neq j, \ \mathcal{C}_i \cap \mathcal{C}_j = \emptyset$$

*where the notion of chain $\mathcal{C}$ is defined in Definition 1 and $\mathcal{C}_i$ denotes the $i^{th}$ chain of $\mathcal{A}_k$.*

*Proof.* We will prove this lemma by induction on $k$. Initialization is straight-forward since, as discussed in Section 3.1, the solution to PAWL$_0$ is such that $\mathcal{A}_0 = \emptyset$.

For the induction step, we now assume that $\mathcal{A}_k$ can be written as the union of disjoint chains. Let $\mathcal{A}_k \cup \{z, z'\}$ be the active set for PAWL$_{k+1}$, and let us assume $z < z'$ without lack of generality. Let us write $\mathcal{A}_k = (\bigcup_i \mathcal{C}_i^{\text{out}}) \bigcup (\bigcup_i \mathcal{C}_i^{\text{in}})$ such that $\{\mathcal{C}_i^{\text{out}}\}$ are chains that lie outside the interval $[z, z']$ and $\{\mathcal{C}_i^{\text{in}}\}$ are chains that lie inside this interval. Since the mappings for $\mathcal{A}_{k+1}$ are those of a 1d OT problem, all the mappings involving points from $\{\mathcal{C}_i^{\text{out}}\}$ will be preserved.

Let us now prove that $(\bigcup_i \mathcal{C}_i^{\text{in}}) \bigcup \{z, z'\}$ is a chain (which would allow us to prove the induction step). Let us assume, by contradiction, that there exists $y \in \mathbf{z}$ such that $z < y < z'$ and $y \notin \mathcal{A}_k$. More precisely, if there exists several such samples, we will take $y$ as the smallest sample satisfying this double inequality. Let us assume that $y$ comes from the same distribution as $z'$ (the case where $y$ comes from the same distribution as $z$ can be derived as a simple symmetry of this one). We will now show that $\mathcal{A}_k \cup \{z, z'\}$ has a higher cost than $\mathcal{A}_k \cup \{z, y\}$ and hence cannot be in the active set at step $k + 1$.

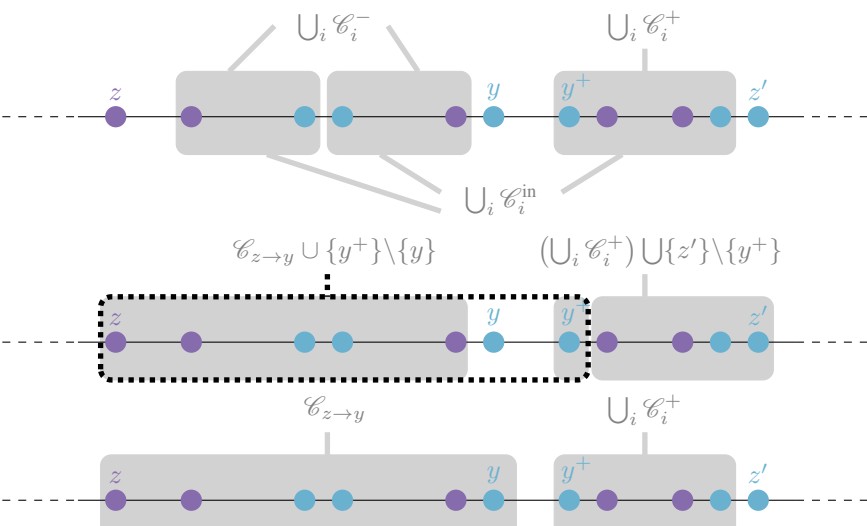

Figure 8: Illustration of the setup considered in the proof for Lemma 1. Here, all points in shaded areas are in $\mathcal{A}_k$ and points in non-shaded areas are in $\overline{\mathcal{A}_k}$. We show in this proof that we cannot have this setup if $\{z, z'\} \in \mathcal{A}_{k+1}$ (top), since $\mathcal{A}_k \cup \{z, y\}$ (middle) would have a lower cost than $\mathcal{A}_k \cup \{z, z'\}$ (bottom).

By the definition of chains, $y$ cannot belong to a chain (as a chain is a set of *consecutive* points), hence $(\bigcup_i \mathcal{C}_i^{\text{in}})$ can be further partitioned into $(\bigcup_i \mathcal{C}_i^-)$ the union of chains whose samples $g$ are such that $g < y$ and $(\bigcup_i \mathcal{C}_i^+)$ the union of chains such that their samples $g > y$. By definition of $y$, we know that $(\bigcup_i \mathcal{C}_i^-) \bigcup \{z, y\}$ forms a chain that we denote $\mathcal{C}_{z \to y}$, hence the cost associated to $\mathcal{A}_k \cup \{z, y\}$ writes:

$$c\left(\bigcup_i \mathcal{C}_i^{\text{out}}\right) + c\left(\bigcup_i \mathcal{C}_i^+\right) + c\left(\mathcal{C}_{z \to y}\right) . \tag{3}$$

On the other hand, let us examine the mappings (and induced cost) that $\mathcal{A}_k \cup \{z, z'\}$ would induce. First, as for the previous case, the mappings related to $(\bigcup_i \mathcal{C}_i^{\text{out}})$ are left unchanged. Let us denote by

$y^+$ the immediate successor of $y$ to be in $\left(\bigcup_i \mathscr{C}_i^+\right)$. Then the cost associated to $\mathcal{A}_k \cup \{z, z'\}$ writes:

$$c\left(\bigcup_i \mathscr{C}_i^{\text{out}}\right) + c\left(\left(\bigcup_i \mathscr{C}_i^+\right)\bigcup\{z'\}\backslash\{y^+\}\right) + c\left(\mathscr{C}_{z\to y} \cup \{y^+\}\backslash\{y\}\right) \tag{4}$$

in order to have the same number of samples from $\boldsymbol{x}$ and $\boldsymbol{y}$ in each part of the cost.

If we first focus on $\mathscr{C}_{z\to y} \cup \{y^+\}\backslash\{y\}$, we see that the mappings in this set will be the same as the mappings in $\mathscr{C}_{z\to y}$ except for the last one that will involve $y^+$ in place of $y$. As $y^+ > y$, we get

$$c\left(\mathscr{C}_{z\to y} \cup \{y^+\}\backslash\{y\}\right) > c\left(\mathscr{C}_{z\to y}\right) . \tag{5}$$

If we now turn our focus on $\left(\left(\bigcup_i \mathscr{C}_i^+\right)\bigcup\{z'\}\backslash\{y^+\}\right)$. the mappings here are all be different from those in $\left(\bigcup_i \mathscr{C}_i^+\right)$. However, given that $z' \notin \mathcal{A}_k$, both these sets could have been candidates to be included in $\mathcal{A}_k$ without changing all other mappings in $\mathcal{A}_k$. Since $\left(\bigcup_i \mathscr{C}_i^+\right)$ was finally included in $\mathcal{A}_k$, we know by optimality of $\mathcal{A}_k$ that:

$$c\left(\left(\bigcup_i \mathscr{C}_i^+\right)\bigcup\{z'\}\backslash\{y^+\}\right) \geq c\left(\bigcup_i \mathscr{C}_i^+\right) \tag{6}$$

Using Equations (5) and (6) in Equations (3) and (4), we get to the conclusion that $c(\mathcal{A}_k \cup \{z, z'\}) > c(\mathcal{A}_k \cup \{z, y\})$, which is a contradiction. $\qquad\square$

One should note that during the induction step of the proof of this lemma, we have proven the current proposition. $\qquad\square$

**Proposition 3.** *Let $\mathscr{C}_{\ell-p\to\ell}$ be a chain. Then we have*

$$c(\mathscr{C}_{\ell-p\to\ell}) = c(\mathscr{C}_{.\to\ell}) - c(\mathscr{C}_{.\to\ell-p-1}) .$$

*Notation $\mathscr{C}_{.\to\ell}$ denotes the maximal chain with a largest element at position $\ell$, i.e. $\mathscr{C}_{.\to\ell} = \arg\min_{\mathscr{C}_{i\to\ell}} \text{card}\left(\mathscr{C}_{i\to\ell}\right)$ and we use the convention $c(\mathscr{C}_{.\to\ell-p-1}) = 0$ if there is no chain with a largest element at position $\ell - p - 1$.*

*Proof of 3.* For this proof, we will rely on the following lemma and the notion of minimal chain defined in definition 3:

**Lemma 2.** *Let $\mathscr{C}$ be a chain. Then there is a unique (up to the order) sequence of adjacent non overlapping minimal chains such that $\mathscr{C} = \bigcup_i \mathscr{C}_i^\star$. Moreover we have $c(\mathscr{C}) = \sum_i c(\mathscr{C}_i^\star)$.*

*Proof.* The proof for this lemma will be made of three parts:

- existence of the sequence can be proven by recursive construction;

- unicity of the sequence stems from the unicity of minimal chains with a largest element at a given position;

- formula for the cost comes from the fact that the cost for a chain is an OT cost in 1d, hence it is the cost of mapping elements from both distributions in the chain in increasing order, which implies that no mapping will occur across minimal chains forming the series.

Let us start by proving, for any chain $\mathscr{C}_{i\to j}$, the existence of a sequence of adjacent non overlapping minimal chains such that the chain is the union of those minimal chains. We will build this set of minimal chains incrementally. We hence start with an empty set and add the minimal chain $\mathscr{C}_j^\star$ whose largest element is at position $j$ in the set. We know that such a minimal chain exists since there is at least one chain whose largest element is at position $j$, which is $\mathscr{C}_{i\to j}$. Let us assume that the first element of $\mathscr{C}_j^\star$ is at position $\ell_1$. By definition of a minimal chain, $\ell_1 \geq i$, so we have $\mathscr{C}_j^\star \subseteq \mathscr{C}_{i\to j}$. Now, if $\ell_1 = i$, we are done, and if not, we are left to finding a minimal chain partition for the chain $\mathscr{C}_{i\to\ell_1-1}$. We can repeat this process recursively until the minimal chain we find covers the remaining

chain (which will occur since we are building a sequence of chains of decreasing length), and the sequence of minimal chains we have extracted hence covers the chain $\mathscr{C}_{i \to j}$.

Let us now prove the unicity of this partition of a chain into minimal chains. Let us assume that we have two sets of minimal chains that would give partitions for $\mathscr{C}_{i \to j}^{\star}$ :

$$\mathscr{C}_{i \to j} = \bigcup_k \mathscr{C}_k^{\star} = \bigcup_{k'} \mathscr{C}_{k'}^{\star} \, .$$

We further assume, without loss of generality, that the sets are ordered in increasing order of their largest position. Since the union of those minimal chains has its largest element at position $j$ (in order to strictly cover $\mathscr{C}_{i \to j}$), the last minimal chain in each set is the one with the largest element at position $j$ (which is unique by definition). We hence have:

$$\mathscr{C}_k^{\star} = \mathscr{C}_{k'}^{\star} \, .$$

The same argument can be used to prove, by recurrence, that the $p$-th minimal chains in both sets are equal, hence both sets are equal, which proves unicity of the partition.

Finally, let us partition the computation of the cost for a chain $\mathscr{C}$. From the partition we just proved, we have:

$$c(\mathscr{C}) = c\left(\bigcup_i \mathscr{C}_i^{\star}\right) \, .$$

Also, we know that the cost of the chain is the Optimal Transport cost between points involved in the chain, which corresponds to mapping the points in order. Given that minimal chains are balanced by definition, we know that mapping the points in order implies mapping the points within the chains, which gives our claimed result:

$$c(\mathscr{C}) = \sum_i c\left(\mathscr{C}_i^{\star}\right) \, .$$

$\square$

Let us now come back to proving that computation for the cost of any chain can be deduced from the cost of *maximal* chains, i.e. the ones that have the highest cardinality.

Let $\mathscr{C}_{\ell-p \to \ell}$ be a chain. We will distinguish between two cases:

Case 1: There is at least one chain whose largest element is at position $\ell - p - 1$. Let us denote by $\ell'$ the smallest index of the maximal chain whose largest element is at position $\ell - p - 1$. We will show that $\ell'$ is also the smallest index of the maximal chain whose largest element is at position $\ell$.

First, let us observe that $\mathscr{C}_{\ell' \to \ell}$ is indeed a chain. This is because $\mathscr{C}_{\ell' \to \ell-p-1}$ and $\mathscr{C}_{\ell-p \to \ell}$ are chains, hence in $\mathscr{C}_{\ell' \to \ell} = \mathscr{C}_{\ell' \to \ell-p-1} \cup \mathscr{C}_{\ell-p \to \ell}$ there are as many samples coming from $\mathbf{x}$ and $\mathbf{y}$.

We now need to prove that $\mathscr{C}_{\ell' \to \ell}$ is maximal. Let us assume, by contradiction, that there exists an index $\ell'' < \ell'$ such that $\mathscr{C}_{\ell'' \to \ell}$ is a chain. Since $\mathscr{C}_{\ell-p \to \ell}$ is a chain, simply counting $\mathbf{x}$ and $\mathbf{y}$ samples in $\mathscr{C}_{\ell'' \to \ell-p-1}$ tells us that it is a chain. We have hence found a chain ending in $\ell - p - 1$ whose first index is strictly smaller than $\ell'$, which contradicts our initial hypothesis about the maximality of $\mathscr{C}_{\ell' \to \ell-p-1}$.

Overall, we have

$$\mathscr{C}_{\cdot \to \ell} = \mathscr{C}_{\ell' \to \ell}$$

and

$$\mathscr{C}_{\cdot \to \ell-p-1} = \mathscr{C}_{\ell' \to \ell-p-1} \, .$$

From Lemma 2, and using the fact that $\mathscr{C}_{\ell' \to \ell-p-1}$, $\mathscr{C}_{\ell' \to \ell}$ and $\mathscr{C}_{\ell-p \to \ell}$ are chains, we have:

$$
\begin{aligned}
c(\mathscr{C}_{\cdot \to \ell}) &= c(\mathscr{C}_{\ell' \to \ell}) \\
&= c(\mathscr{C}_{\ell' \to \ell-p-1}) + c(\mathscr{C}_{\ell-p \to \ell}) \\
&= c(\mathscr{C}_{\cdot \to \ell-p-1}) + c(\mathscr{C}_{\ell-p \to \ell})
\end{aligned}
$$

Case 2: There is no chain ending at position $\ell - p - 1$. In this case, we have $c(\mathscr{C}_{\cdot \to \ell-p-1}) = 0$ by definition, hence we need to prove $c(\mathscr{C}_{\ell-p \to \ell}) = c(\mathscr{C}_{\cdot \to \ell})$. Let us assume, by contradiction, that there

exists a chain $\mathscr{C}_{\ell' \to \ell}$ such that $\ell' < \ell - p$. We can hence use Lemma 2 to partition this chain in a set of contiguous non-overlapping minimal chains. Since $\mathscr{C}_{\ell-p \to \ell}$ is a chain too, we know that the partition of $\mathscr{C}_{\ell' \to \ell}$ can be written:

$$\mathscr{C}_{\ell' \to \ell} = \underbrace{\left( \bigcup_i \mathscr{C}_i^\star \right)}_{\mathscr{C}_{\ell' \to \ell - p - 1}} \cup \underbrace{\left( \bigcup_i \mathscr{C}_i^\star \right)}_{\mathscr{C}_{\ell - p \to \ell}}$$

and this contradicts our initial hypothesis that there is no chain ending at position $\ell - p - 1$. There is hence no chain ending at position $\ell$ that starts before $\ell - p$ and we have $\mathscr{C}_{\ell-p \to \ell} = \mathscr{C}_{\cdot \to \ell}$, from which the cost equality follows. $\qquad \square$

**Proposition 4.** *Assuming that $\boldsymbol{z}$ is sorted, extracting all minimal chains can be done in $\mathcal{O}\left(n\right)$ by iterating over items in $\boldsymbol{z}$ in increasing order.*

*Proof of 4.* Let us start by extracting all minimal chains from the sorted union distribution $\boldsymbol{z}$. To do so, we will rely on the notion of differential rank, defined below and illustrated in Figure 2:

**Definition 4.** *The differential rank at position $\ell$, denoted $r_\ell$, is defined as*
$$r_\ell = Card(\{x \in \mathbf{x} | x \le z_\ell\}) - Card(\{y \in \mathbf{y} | y \le z_\ell\}).$$
*By convention, we set $r_0 = 0$.*

Differential ranks can be computed in $\mathcal{O}\left(n\right)$ given sorted $\boldsymbol{z}$, and they can later be used to extract all chains from $\boldsymbol{z}$, given the following property:

**Lemma 3.** *The following two statements are equivalent:*

1. *$\mathscr{C}_{\ell-p \to \ell}^\star$ is a minimal chain in $\boldsymbol{z}$*

2. *$p$ is the smallest strictly positive integer such that $r_\ell = r_{\ell-p-1}$*

*Proof.* Let us start by proving $1. \Rightarrow 2.$ Let $\mathscr{C}_{\ell-p \to \ell}^\star$ be a minimal chain in $\boldsymbol{z}$. Then we have:

$$
\begin{aligned}
r_\ell &= Card(\{x \in \boldsymbol{x} | x \le z_\ell\}) - Card(\{y \in \boldsymbol{y} | y \le z_\ell\}) \\
&= (Card(\{x \in \boldsymbol{x} | x \le z_{\ell-p-1}\}) + Card(\{x \in \boldsymbol{x} | x \in \mathscr{C}_{\ell-p \to \ell}\})) \\
&\quad - (Card(\{y \in \boldsymbol{y} | y \le z_{\ell-p-1}\}) + Card(\{y \in \boldsymbol{y} | y \in \mathscr{C}_{\ell-p \to \ell}\})) \\
&= r_{\ell-p-1} + \underbrace{Card(\{x \in \boldsymbol{x} | x \in \mathscr{C}_{\ell-p \to \ell}\}) - Card(\{y \in \boldsymbol{y} | y \in \mathscr{C}_{\ell-p \to \ell}\})}_{0 \text{ since } \mathscr{C}_{\ell-p \to \ell} \text{ is a chain}}
\end{aligned}
$$

Now we still have to prove that $p$ is the smallest strictly positive integer such that $r_\ell = r_{\ell-p-1}$, which we will do by contradiction. Let us assume that there exists a strictly positive integer $p'$ such that $\ell - p < \ell - p' < \ell$ and $r_{\ell-p'-1} = r_\ell$. Then we have:

$$
\begin{aligned}
r_{\ell-p'-1} = r_\ell \quad &\Leftrightarrow \quad Card(\{x \in \boldsymbol{x} | x \le z_{\ell-p'-1}\}) - Card(\{y \in \boldsymbol{y} | y \le z_{\ell-p'-1}\}) \\
&\qquad = Card(\{x \in \boldsymbol{x} | x \le z_\ell\}) - Card(\{y \in \boldsymbol{y} | y \le z_\ell\}) \\
&\Leftrightarrow \quad Card(\{x \in \boldsymbol{x} | x \le z_\ell\}) - Card(\{x \in \boldsymbol{x} | x \le z_{\ell-p'-1}\}) \\
&\qquad = Card(\{y \in \boldsymbol{y} | y \le z_\ell\}) - Card(\{y \in \boldsymbol{y} | y \le z_{\ell-p'-1}\}) \\
&\Leftrightarrow \quad Card(\{x \in \boldsymbol{x} | z_{\ell-p'} \le x \le z_\ell\}) = Card(\{y \in \boldsymbol{y} | z_{\ell-p'} \le y \le z_\ell\})
\end{aligned}
$$

$\mathscr{C}_{\ell-p' \to \ell}$ is hence a chain, which contradicts the minimality of $\mathscr{C}_{\ell-p \to \ell}^\star$.

Let us now prove $2. \Rightarrow 1.$ We hence assume that $p$ is the smallest strictly positive integer such that $r_\ell = r_{\ell-p-1}$. Using the series of equivalences above, we have that

$$Card(\{x \in \boldsymbol{x} | z_{\ell-p} \le x \le z_\ell\}) = Card(\{y \in \boldsymbol{y} | z_{\ell-p} \le y \le z_\ell\})$$

hence $\mathscr{C}_{\ell-p \to \ell}$ is a chain. We now need to prove that this is the smallest chain whose smallest element is at position $\ell$, which we will do, once again, by contradiction. Let us assume that there exists $\ell'$ such that $\ell - p < \ell' < \ell$ and $\mathscr{C}_{\ell' \to \ell}$ is a chain. Then using the series of equivalences above once again, we get that $r_{\ell'-1} = r_{\ell-p-1} = r_\ell$, which contradicts the fact that $p$ is the smallest strictly positive integer such that $r_\ell = r_{\ell-p-1}$. $\qquad \square$

We will now rely on this lemma to efficiently extract all minimal chains by iterating over items in $z$ in increasing order. Since a chain contains as many $x$'s as $y$'s by definition, it should preserve the differential ranks (i.e. the differential rank before the chain should be equal to that after the chain).

Given this property, extracting all chains from $z$ can be done in $\mathcal{O}(n)$ by iterating over positions in increasing order: at each position $\ell$, it is sufficient to compute $r_\ell$ based on $r_{\ell-1}$ and $z_\ell$. Then, a lookup table storing the last occurrence of each differential rank value can be used to decide if a chain has a largest element at the current position and, if so, get the smallest position of the chain.

$\square$

**Proposition 5.** *Let us denote the cumulative sums* $C^{\mathbf{x}}_{\rightarrow \ell} = \sum_{i \le \ell} z_i \mathbb{1}_{z_i \in \mathbf{x}}$ *and similarly* $C^{\mathbf{y}}_{\rightarrow \ell} = \sum_{i \le \ell} z_i \mathbb{1}_{z_i \in \mathbf{y}}$, *with* $C^{\mathbf{x}}_{\rightarrow 0} = C^{\mathbf{y}}_{\rightarrow 0} = 0$ *by convention. The Manhattan cost for the minimal chain* $\mathscr{C}^{\star}_\ell = \{z_{\ell-p}, \dots, z_\ell\}$ *is:*

$$c(\mathscr{C}^{\star}_\ell) = \left| C^{\mathbf{x}}_{\rightarrow \ell} - C^{\mathbf{y}}_{\rightarrow \ell} - C^{\mathbf{x}}_{\rightarrow \ell-p-1} + C^{\mathbf{y}}_{\rightarrow \ell-p-1} \right|$$

*which can be computed in* $\mathcal{O}(1)$ *time once the cumulative sums stored.*

*Proof of 5.* We have:

$$c(\mathscr{C}^{\star}_\ell) = |z_{\ell-p} - z_{\ell-p+1}| + |z_{\ell-p+2} - z_{\ell-p+3}| + \cdots + |z_{\ell-1} - z_\ell|$$

Since inside a minimal chain all mappings have the same ordering (shown in Lemma 4 below), then all the quantities inside $|\cdot|$ have the same sign, hence

$$
\begin{aligned}
c(\mathscr{C}^{\star}_\ell) &= |z_{\ell-p} - z_{\ell-p+1} + z_{\ell-p+2} - z_{\ell-p+3} + \cdots + z_{\ell-1} - z_\ell| \\
&= |(z_{\ell-p} + z_{\ell-p+2} + \cdots + z_{\ell-1}) - (z_{\ell-p+1} + z_{\ell-p+3} + \cdots + z_\ell)| \\
&= \left| \sum_{k=\ell-p}^{\ell} z_k \mathbb{1}_{z_k \in \mathbf{x}} - \sum_{k=\ell-p}^{\ell} z_k \mathbb{1}_{z_k \in \mathbf{y}} \right| \\
&= \left| \left( \sum_{k \le \ell} z_k \mathbb{1}_{z_k \in \mathbf{x}} - \sum_{k \le \ell-p-1} z_k \mathbb{1}_{z_k \in \mathbf{x}} \right) - \left( \sum_{k \le \ell} z_k \mathbb{1}_{z_k \in \mathbf{y}} - \sum_{k \le \ell-p-1} z_k \mathbb{1}_{z_k \in \mathbf{y}} \right) \right| \\
&= \left| C^{\mathbf{x}}_{\rightarrow m} - C^{\mathbf{y}}_{\rightarrow m} - C^{\mathbf{x}}_{\rightarrow l-1} + C^{\mathbf{y}}_{\rightarrow l-1} \right|
\end{aligned}
$$

**Lemma 4.** *Inside a minimal chain, all mappings have the same ordering, i.e. either* $x_i < y_i$, $\forall x_i, y_i \in \mathscr{C}^\star$ *or* $x_i > y_i$, $\forall x_i, y_i \in \mathscr{C}^\star$.

*Proof.* Let $\mathscr{C}^{\star}_\ell = \{z_{\ell-p}, \dots, z_\ell\}$ be a minimal chain. Let us assume, without loss of generality, that $z_{\ell-p} \in \mathbf{x}$. We will now show that all samples from $\mathbf{x} \cap \mathscr{C}^{\star}_\ell$ are mapped to samples from $\mathbf{y} \cap \mathscr{C}^{\star}_\ell$ that are greater than them. By contradiction, let us assume that there is at least one mapping inside $\mathscr{C}^{\star}_\ell$ that goes the other way around. Let us denote by $\hat{x}$ and $\hat{y}$ the samples involved in the leftmost of such mapping on the line. We hence have $\hat{y} < \hat{x}$.

By definition of $\{\hat{x}, \hat{y}\}$, all $\mathbf{y}$ samples in $\mathscr{C}^{\star}_\ell$ that are lower than $\hat{y}$ are mapped to $\mathbf{x}$ samples that are lower than themselves. This means that

$$\text{card}(x \in \mathbf{x} \cap \mathscr{C}^{\star}_\ell | x < \hat{y}) \ge \text{card}(y \in \mathbf{y} \cap \mathscr{C}^{\star}_\ell | y < \hat{y}).$$

Similarly, since samples are matched in order on the real line, all $\mathbf{x}$ samples in $\mathscr{C}^{\star}_\ell$ that are lower than $\hat{x}$ have to be mapped to a sample that is lower than $\hat{y}$, and we get

$$\text{card}(x \in \mathbf{x} \cap \mathscr{C}^{\star}_\ell | x < \hat{y}) \le \text{card}(y \in \mathbf{y} \cap \mathscr{C}^{\star}_\ell | y < \hat{y}).$$

By combining these two inequalities, the set of points

$$\{z \in \mathscr{C}^{\star}_\ell | z < \hat{y}\}$$

is a balanced set of contiguous points, hence it is a chain, which contradicts the minimality of $\mathscr{C}^{\star}_\ell$. $\square$

$\square$

**Proposition 6.** *Sliced-PW$(\mu, \nu, s)$ is an upper bound of PW and it is a semi-metric almost surely.*

*Proof of 6.* Let us first recall that

$$\text{Sliced-PW}(\mu, \nu, s) = \min_{\theta \in \mathbb{S}^d} \sum_{i,j} d(\boldsymbol{x}_i, \boldsymbol{y}_j) \pi_{ij}^{\star}(\theta) \text{ with } \boldsymbol{\pi}^{\star}(\theta) = \arg\min_{\boldsymbol{\pi} \in \Pi(\mu_{\leq}, \nu_{\leq})} \sum_{i,j} d(\langle \boldsymbol{x}_i, \theta \rangle, \langle \boldsymbol{y}_j, \theta \rangle) \pi_{ij}$$

In brief, Sliced-PW$(\mu, \nu, s)$ is a semi-metric as long as $\langle \boldsymbol{x}_i, \theta \rangle \neq \langle \boldsymbol{x}_{i'}, \theta \rangle, \forall i \neq i'$, which is true almost surely for distributions living in $\mathbb{R}^d$. This requirement is also needed for using PAWL as the latter requires the distributions to be disjointly supported.

*Upper bound.* It is easy to see that $\text{PWL}(\mu, \nu, s) \leq \text{Sliced-PW}(\mu, \nu, s)$ as the optimal transport matrices of the two problems live in the same set of constraints and that the cost matrices are the same.

*Non negativity.* As the cost is a distance $d$, Sliced-PW$(\mu, \nu, s) \geq 0$.

*Symmetry.* It can be easily shown as by noticing that $d(\langle \boldsymbol{x}_i, \theta \rangle, \langle \boldsymbol{y}_j, \theta \rangle) = d(\langle \boldsymbol{y}_j, \theta \rangle, \langle \boldsymbol{x}_i, \theta \rangle)$.

*Identity.* Let us first consider that $\nu = \mu$. It implies that $\boldsymbol{x}_i = \boldsymbol{y}_i, \forall i$, hence $d(\boldsymbol{x}_i, \boldsymbol{y}_i) = 0 \Leftrightarrow d(\langle \boldsymbol{x}_i, \theta \rangle, \langle \boldsymbol{y}_i, \theta \rangle) = 0, \forall \theta$. We thus have Sliced-PW$(\mu, \mu, s) = 0, \forall s, \theta$.

Let us now suppose that Sliced-PW$(\mu, \nu, s) = 0$. Note $\theta^{\star}$ the optimal direction. It means that $d(\boldsymbol{x}_i, \boldsymbol{y}_j) \pi_{ij}^{\star}(\theta^{\star}) = 0, \forall i, j$. We first consider problem PAWL$_k$, with $k = \lfloor \frac{s}{w} \rfloor$.

Let us denote $\mu^k$ the active support of $\mu$ and $\nu^k$ the active support of $\nu$. We straighforwardly have Sliced-PW$(\mu, \nu, k \cdot w) = \text{Sliced-PW}(\mu^k, \nu^k, k \cdot w) = 0$. As $a_i = b_j = w$, $\pi_{ij}^{\star}(\theta^{\star})$ is a permutation matrix. We define the injective map $f : \{1, 2, \cdots, k\} \rightarrow \{1, 2, \cdots, k\}$ such that $\text{PAWL}_k = \sum_i d(\boldsymbol{x}_i, \boldsymbol{y}_{f(i)})$. As we suppose that Sliced-PW$(\mu, \nu, k \cdot w) = 0$, it means that $\text{PAWL}_k = \sum_{i=1}^k d(\boldsymbol{x}_i, \boldsymbol{y}_{f(i)}) = 0 \Leftrightarrow \boldsymbol{x}_i = \boldsymbol{y}_{f(i)}, \forall i$, but also that $\sum_i d(\langle \boldsymbol{x}_i, \theta^{\star} \rangle, \langle \boldsymbol{y}_{f(i)}, \theta^{\star} \rangle) = \langle d(\boldsymbol{x}_i - \boldsymbol{y}_{f(i)}), \theta^{\star} \rangle = 0$. In addition, as $\theta^{\star}$ belongs to the unit sphere, and with the assumption of disjointly supported distributions on the line, it holds true iif $\boldsymbol{x}_i = \boldsymbol{y}_{f(i)}$.

We then have $\mu_k = \nu_k$ when Sliced-PW$(\mu, \nu, k \cdot w) = 0$. The extension of this result for PWL$(s)$ comes directly from Prop. 1 and is omitted here.

**Discussion about the triangular inequality.** We claim here that, despite having a metric is of prime importance for assessing weak convergence for instance, it may be an undesirable property in case of Partial Wasserstein. To illustrate our point, just consider the following scenario: $\mu$ is a bimodal distribution with one mode of mean $m_\mu^1 = -2$ and $m_\mu^2 = 2$ (where most of the samples are sampled according to the first mode), $\nu$ is another bimodal distribution with one mode of mean $m_\nu^1 = -2$ and $m_\nu^2 = 2$ (where most of the samples are sampled according to the second mode) and third distribution $\alpha$ with same amount of mass on the two modes. It is easy to see that, at least for some amount of mass $s$, the Partial Wasserstein distance between $\mu$ and $\nu$ will be greater than the sum of the Wasserstein distance between $\mu$ and $\alpha$ and $\alpha$ and $\nu$, which makes sense in a machine learning context for instance.

When triangular inequality is sought (e.g. when metrizing the weak convergence), as soon as the partial distributions $\mu^k$ and $\nu^k$ that are at stake are stable over the iterations, one can rely on the metric properties of SWGG. We left investigation of this behavior as a future work.

$\square$

## A.2 COMPUTATIONAL COMPLEXITY

**Proposition 7.** *As claimed in the paper, computational complexity is $\mathcal{O}(n \log n)$ when the cost is the Manhattan distance and $\mathcal{O}(Cn^2)$ for any other cost, where $C$ is the complexity of a single distance computation.*

*Proof.* One should first note that Algorithms 1 and 2 require sorted distributions. Sorting distributions of size $n$ is an $\mathcal{O}(n \log n)$ complexity operation. Let us now focus on the study of the time complexity

---

**Algorithm 3:** Computation of chain costs for costs other than the Manhattan distance

---

**Data:** Sorted $x$, Sorted $y$, Sorted $z$
 ▷ Precomputations (to be performed once and for all)
1 Extract all minimal chains from $z$ ▷ Uses Proposition 4
2 **for** $j \in [1..n]$ **do**
3  **if** $\mathscr{C}_j^\star$ *exists* **then**
4   Compute $c\left(\mathscr{C}_j^\star\right)$
5  **end**
6  Compute $c\left(\mathscr{C}_{.\rightarrow j}\right)$ using Equation (2)
7 **end**

 ▷ Computation of a chain marginal cost using eq. 1 and prop. 3
8 $\text{marginal\_cost}\left(\mathscr{C}_{\ell-p\rightarrow\ell}\right) \leftarrow c\left(\mathscr{C}_{.\rightarrow\ell+1}\right) - c\left(\mathscr{C}_{.\rightarrow\ell-p-1}\right) - c\left(\mathscr{C}_{.\rightarrow\ell}\right) + c\left(\mathscr{C}_{.\rightarrow\ell-p}\right)$

---

of the precomputations presented in Algorithm 1 (for the Manhattan cost) and Algorithm 3 (for other costs).

Cumulative sums that are computed at line 1 of Algorithm 1 can be obtained in $\mathcal{O}(n)$. Then, as stated in Proposition 4, all minimal chains can be extracted from $z$ in $\mathcal{O}(n)$ time.

The loop over $j$ has $n$ iterations, and at each iteration:

- The existence of $\mathscr{C}_j^\star$ can be tested in $\mathcal{O}(1)$ thanks to the extraction of **all** minimal chains performed beforehand;

- For the Manhattan cost, the computation of $c\left(\mathscr{C}_j^\star\right)$ can be done in $\mathcal{O}(1)$ time (cf. Proposition 5)

- For other costs, assuming a single cost computation takes $\mathcal{O}(C)$ time, the computation of $c\left(\mathscr{C}_j^\star\right)$ can be done in $\mathcal{O}\left(C \cdot \text{card}(\mathscr{C}_j^\star)\right)$ time;

- Using Equation (2), computing $c\left(\mathscr{C}_j^\star\right)$ can be done in $\mathcal{O}(1)$.

Overall, the computational complexity for this loop is $\mathcal{O}(n \log n)$ in the Manhattan case and $\mathcal{O}(C \cdot S)$, where $S = \sum_j \text{card}(\mathscr{C}_j^\star)$ for other costs. In practice, $\mathcal{O}(n) \leq \mathcal{O}(S) \leq \mathcal{O}(n^2)$, hence our worst-case complexity for this step is $\mathcal{O}(C \cdot n^2)$ if the cost is not Manhattan. The last step that consists in the marginal cost computation is in all cases $\mathcal{O}(1)$.

If we now study the complexity of Algorithm 2 that operates after the precomputations of Algorithm 1 or Algorithm 3, we can first notice that this algorithm will have the same complexity whatever the cost, since all required costs computations have been performed beforehand. The first step in this algorithm is to initialize the list of candidates with all candidate pairs that consist of neighbors in $z$. As discussed in Section 3.3 (S1 case), this can be done in $\mathcal{O}(n)$ time. Since we rely here on a heap queue implementation of the list of candidates in order to keep it sorted in increasing order of the marginal costs, the cost is extended to $\mathcal{O}(n \log n)$. Then, at each of the $\lceil \frac{s}{w} \rceil$ steps of the algorithm, the minimal element is popped from list_candidates (this is an $\mathcal{O}(1)$ operation thanks to our heap-queue-based sorted list of candidates). Extracting the covering chain of the current samples can be done in $\mathcal{O}(1)$ time, as long as one maintains a dictionary of maximal chains in the active set throughout the execution of the algorithm.

In practice, cleaning could be a costly operation, and we decide to postpone it. In other words, at each step $k$, we need to check whether sample $x_i$ or $y_j$ is already included in $\mathcal{A}_{k-1}$: if so, the iteration is skipped.

Finally, insertion in a sorted list implemented by a heap queue is $\mathcal{O}(\log n)$ given that the list is of size at most $n$. Overall, execution of this algorithm runs in $\mathcal{O}(n \log n)$ time.

Finally, the full execution of our PAWL solver algorithm takes:

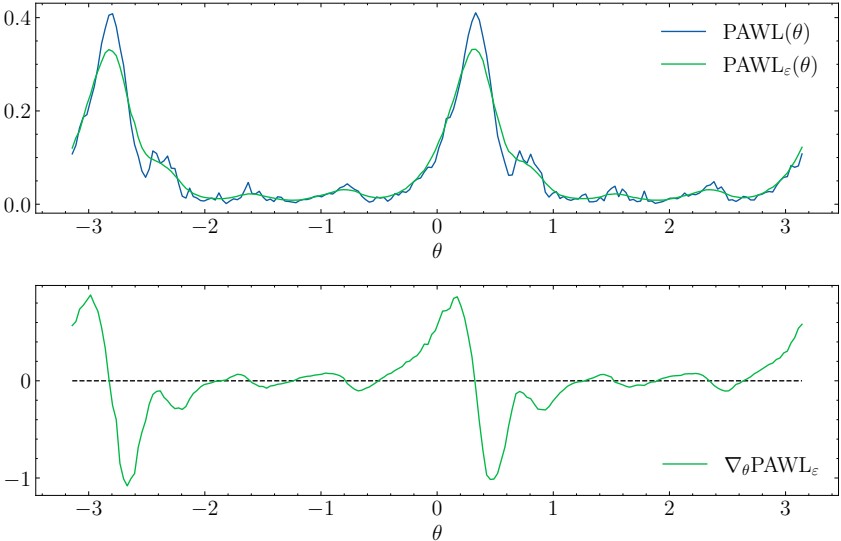

Figure 9: Perturbed PAWL (top) and its gradient (bottom).

- $\mathcal{O}\left(n \log n\right)$ time if the Manhattan cost is used;

- $\mathcal{O}\left(Cn^2\right)$ time, or $\mathcal{O}\left(CnS + n \log n\right)$ (where $2S$ is the size of the largest minimal chain) for a tighter estimation, otherwise.

$\square$

### A.3 PERTURBED PAWL

As noted in the literature (e.g., (Blondel et al., 2020)), back-propagating through sorting-based algorithms like PAWL poses challenges due to the non-smooth nature of the operations involved. Specifically, if PAWL is treated as a function of some parameter $\theta$, its gradient can be expressed as:

$$\nabla_\theta \text{PAWL} = \left\langle \pi^\star, \frac{\partial C}{\partial \theta} \right\rangle$$

where $\pi^\star$ is the optimal transport plan and $C$ is the cost matrix. However, this gradient is often unstable, making it unsuitable for direct use in standard gradient-based optimization methods. To address this, we can leverage perturbed optimizers (e.g., Berthet et al. (2020)), which introduce noise to smooth the optimization landscape.

For illustration, consider a 2D setup where we compute the PAWL distance between two distributions $\mu$ and $\nu$, projected onto 1D as a function of the orientation $\theta$ of the projection direction. Using a perturbed optimizer, the smoothed objective $\text{PAWL}_\epsilon(\theta)$ is defined as:

$$\text{PAWL}_\epsilon(\theta) = \mathbb{E}_{z \sim \mathcal{N}_{0,1}} \left[\text{PAWL}(\theta + \epsilon z)\right]$$

where $epsilon$ controls the magnitude of the perturbation. One can then use Stein's lemma to get the following expression for its gradient (cf. Blondel & Roulet (2024), Section 14.4.5):

$$\nabla_\theta \text{PAWL}_\epsilon = \mathbb{E}_{z \sim \mathcal{N}_{0,1}} \left[\left(\text{PAWL}(\theta + \epsilon z) - \text{PAWL}(\theta)\right) \cdot \frac{z}{\epsilon}\right]$$

Figure 9 illustrates this approach, showing the smoothed PAWL (top) and its gradient (bottom) with Monte Carlo estimation using 1k samples and $\epsilon = 0.1$. As shown, this perturbation is sufficient to smooth the gradients, making them more stable and suitable for gradient-based optimization in neural network contexts.

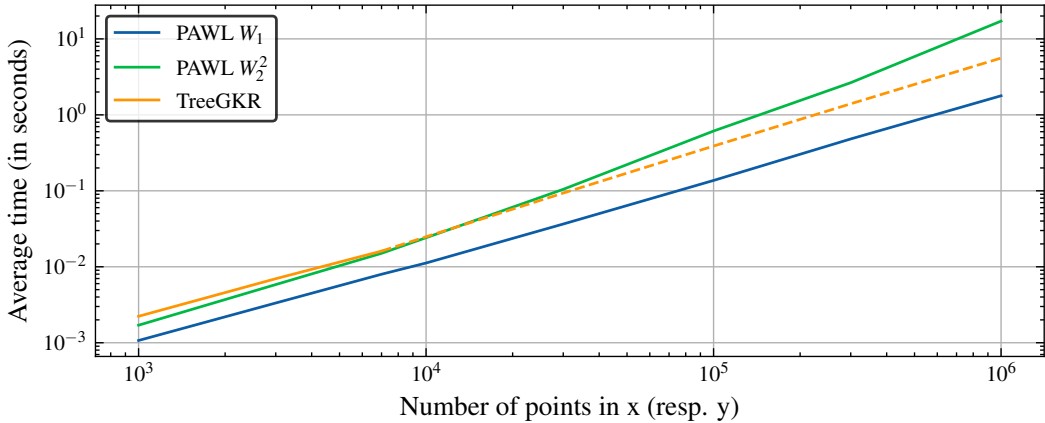

Figure 10: Running time comparisons between our PAWL solvers and TreeGKR. Dashed line corresponds to extrapolation based on theoretical complexity.

## A.4 ADDITIONAL EXPERIMENTS

### A.4.1 INCLUDING SATO ET AL. (2020) IN THE TIMINGS COMPARISON

TreeGKR (Sato et al., 2020) is not included in our benchmark of the computational complexities of the partial and unbalanced optimal transport methods, though it constitutes a serious competitor, since a 1D unbalanced OT problem can be cast to a tree OT problem. Indeed, for samples on the real line, one can build a path graph (which is a specific kind of tree) such that the distances in 1D are reflected on the tree.

Unfortunately, we were unable to reuse code from (Sato et al., 2020) on large datasets. As a consequence, we present in Figure 10 a benchmark that involves TreeGKR for sample size up to 7k. For larger sample sizes, the complexity is extrapolated using the theoretical $\mathcal{O}\left(n \log^2 n\right)$ complexity for TreeGKR, for the sake of visualization.

### A.4.2 IMPACT OF THE TRANSPORTED MASS ON THE MEASURED TIMINGS

In Figure 4, we provide running times for PAWL compared with two baselines, namely OPT (Bai et al., 2023) and Fast-UOT (Séjourné et al., 2022). Partial transportation problems for all intermediate masses are solved at once for PAWL, but for the baselines, the amount of transported mass is controlled via a hyper-parameter of the method. Figure 11 shows that varying the $\lambda$ hyper-parameter for OPT does not seriously impact its temporal complexity.[2] Note also that the selected values for $\lambda$ cover a wide range of transported mass $\bar{s}$.

---

[2]Similar observations could be drawn for Fast-UOT when varying the $\rho$ hyper-parameter.

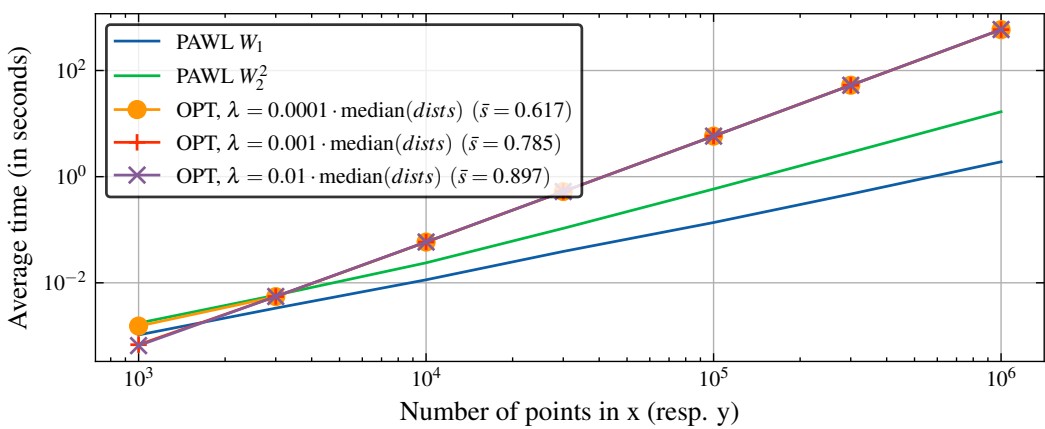

Figure 11: Running time comparisons between our PAWL solvers and OPT with varying $\lambda$. $\bar{s}$ is the average transported mass.

