# OpenReview forum: "One for all and all for one: Efficient computation of partial Wasserstein distances on the line"
_ICLR.cc/2025/Conference — ICLR 2025 Poster_

### Official Review · Reviewer_ErWA · 2024-10-28

**Soundness:** 2
**Presentation:** 2
**Contribution:** 3
**Rating:** 6
**Confidence:** 4

**Summary:**

1. The paper presents a new algorithm for solving the partial Wasserstein problem in a straightforward manner.
2. It introduces the concept of sliced-partial Wasserstein and discusses its semi-metric properties.
3. The new algorithm is applied to shape registration experiments, and an evaluation of its performance is provided.

**Strengths:**

1. The classical solver for the proposed partial optimal transport (OT) problem requires $O(n^3)$ complexity for linear programming or $O(n^2 / \epsilon)$ for the Sinkhorn algorithm. This paper proposes an efficient algorithm specifically for the Manhattan distance in a one-dimensional, empirical distribution setting.
2. The sliced-partial Wasserstein (sliced-PW) concept is introduced, and its semi-metric properties are analyzed.
3. The authors demonstrate the application of this new algorithm in a shape registration experiment.

**Weaknesses:**

1. **Regarding Complexity**:
   - 1.1 If the cost is not based on the Manhattan distance, the computational complexity of the new algorithm is \( k \cdot n^2 \). In the worst-case scenario (when \( k = n \)), this becomes \( n^3 \). Is this correct?
   - 1.2 In Section 5.1 and Figure 4, it seems the authors discuss only cases where the cost is a metric. I recommend the following:
     - (a) Include **(Sato et al., 2020)** as a baseline. The two baselines from **(Bai et al., 2023)** and **(Sejourne et al., 2023)** apply when the cost is not a metric, while Sato's method and the algorithm proposed in this paper are specifically designed for metric costs. In the one-dimensional metric cost scenario, Sato's algorithm appears to have a comparable computational cost of \( n \log^2 n \).
     - (b) Upon reviewing the anonymized repository (specifically `timing_partial.py`), it seems that only the extreme case of full mass transportation has been tested. I recommend measuring the wall-clock time for various mass transport settings, such as transporting 20%, 50%, and 100% of the total mass.
   - 1.3 In Section 5.1, the parameters for fast UOT are set as \( \rho_1 = \rho_2 = 1.0 \), and for OPT, \( \lambda = 0.001 \times \text{median} \). However, it appears that the parameter for PAWL (denoted as \( s \)) is missing, as I could not find it in the code repository. Could you clarify the missing parameter and explain why the resulting problems are equivalent or comparable under these settings?

2. **Regarding the Semi-Metric Properties of Sliced-PW**: When \( \text{sliced-PW}(\mu, \nu; k \cdot \omega) = 0 \), it implies \( \mu^k = \nu^k \), but we cannot conclude that \( \mu = \nu \) (see lines 998–1002 in the appendix). Therefore, sliced-PW does not satisfy the triangle inequality or identity properties, though it does exhibit symmetry and non-negativity. It appears that sliced-PW was not applied in the experiments, and its potential applications were neither discussed nor demonstrated.


Minor comments:
It appears that the proposed algorithm primarily addresses cases where $\mu$ and $\nu$ are empirical measures with equal mass on each point. If this is the case, setting $\omega = 1$ makes the equations for (PAWL) and classical partial OT, i.e., Eq (PW), equivalent. Therefore, introducing the new problem (PAWL)  seems unnecessary, as it merely changes the notation of the original partial OT problem. (This comment is only about notation redundancy and does not affect my overall rating.)

**Questions:**

1. In the 1D case, what is the difference between "Manhattan distance" and basic absolute distance?
2. In the repository `repo/baselines/cvpr_2023_bai.icp_xp.py`, it appears that all the baseline methods are not accelerated by Numba; however, Numba acceleration is implemented in the original repository at [yikun-baio/sliced_opt](https://github.com/yikun-baio/sliced_opt). Could you test the convergence time using the original code from [yikun-baio/sliced_opt](https://github.com/yikun-baio/sliced_opt)?
3. Can you provide a formal proposition/theorem (with a formal proof) to present the proposed solver's computational cost for solving a partial Wasserstein problem, both with and without the Manhattan metric assumption?
4. Could you print the transportation plan for the new algorithm in the following partial OT problem?

Let \(\mu = \delta_{x_1} + \delta_{x_2} + \delta_{x_3}\), where \(x_1 = 0\), \(x_2 = 1\), \(x_3 = 2\) (with support on points 0, 1, 2, and each point having a mass of 1), and \(\nu = \delta_{y_1} + \delta_{y_2} + \delta_{y_3}\), where \(y_1 = 2\), \(y_2 = 3\), \(y_3 = 4\) (with support on points 2, 3, 4, and each point having a mass of 1).

The transported mass is set to \(k = s = 3\), with \(w = 1\). The cost function can be either the Manhattan distance or the square of the Manhattan distance. When using the squared distance, the optimal transportation plan is \(x_1 \rightarrow y_1\), \(x_2 \rightarrow y_2\), and \(x_3 \rightarrow y_3\). I am curious to see the outcome of this new algorithm.

---

> ### Author Response · Authors · 2024-11-26
>
> Following our response, we want to ensure that the clarifications we provided adequately address your questions and comments. If any aspect remains unclear or requires additional explanation, please do not hesitate to let us know. We would be happy to provide further details or rephrase certain parts to ensure that all your concerns are fully addressed. In particular, your question regarding Numba was highly relevant, and we would like to ensure that our response has resolved any ambiguity.
> Once again, we sincerely appreciate your valuable contribution to improving our manuscript.

---

> > ### Comment · Reviewer_ErWA · 2024-12-02
> >
> > > Could you print the transportation plan for the new algorithm in the following partial OT problem?
> >
> > It seems my question wasn’t fully addressed.
> >
> > Could you present the transportation plan in every iteration of your new algorithm for this simple 1-dimensional partial OT problem?
> > - $\mu$ is an empirical distribution supported on points $0, 1, 2$.
> > - $\nu$ is an empirical distribution supported on points $-2, -1, 0$.
> > - $k = s = 3$.  Manhattan distance
> >
> > I would like to know the transportation plan at every iteration. Could you verify whether the transportation plan in your algorithm (from $\mu$ to $\nu$) in each iteration matches the following?
> > - **Iteration 1**: $0 \to 0$
> > - **Iteration 2**: $0 \to -1, 1 \to 0$
> > - **Iteration 3**: $0 \to -2, 1 \to -1, 2 \to 0$
> >
> > If this is correct, I’m confused by the statement (Proposition 7) that the complexity is $O(n \ln(n))$. Even if all other parts of the computational cost were $0$, updating the transportation plan in iteration $k$ requires $k$ computations. This suggests that the computational cost would be $1 + 2 + \dots + n = O(n^2)$.
> >
> > Could you clarify this?

---

> > > ### Author Response · Authors · 2024-12-02
> > >
> > > We sincerely appreciate your detailed inquiry, which allowed us to clarify the process and complexities associated with our algorithm.
> > >
> > > In this answer, we first unroll the execution of our algorithm on your provided example and then discuss the specific point of recovering *transportation plans* from the outcomes of our algorithm (namely transportation costs and active sets).
> > >
> > > ### Execution of our algorithm on your example
> > >
> > >
> > > In this specific example, distribution $z = \mu \cup \nu$ is the distribution supported on samples $-2,-1,0,0,1,2$ (sorting $z$ can be done in $O(n \log n)$).
> > > Sorted $z$ is hence: $[\mu_1]$ -- $[\mu_2]$ -- $[\nu_1]$ -- $[\mu_3]$ -- $[\nu_2]$ -- $[\nu_3]$ (note, because $\mu_3 = \nu_1$, we could also consider $[\mu_1]$ -- $[\mu_2]$ -- $[\mu_3]$ -- $[\nu_1]$ -- $[\nu_2]$ -- $[\nu_3]$ without any changes on the output of the algorithm).
> > >
> > > The first step consists in precomputations as described in Algorithm 1.
> > > The minimal chains in this problems are (note that they are indexed by the position of their right-most element in $z$ and indices start at one):
> > > - $\mathcal{C}^\star_3 = {\mu_2, \nu_1}$ of $L_1$ cost 1
> > > - $\mathcal{C}^\star_4 = {\nu_1, \mu_3}$ of $L_1$ cost 0
> > > - $\mathcal{C}^\star_5 = {\mu_3, \nu_2}$ of $L_1$ cost 1
> > > - $\mathcal{C}^\star_6 = {\mu_1, \mu_2, \nu_1, \mu_3, \nu_2, \nu_3}$ of $L_1$ cost 6
> > >
> > > Then for all positions, we can compute the cost of the longest chain ending at that position, using dynamic programming:
> > > - $\text{cost} (\mathcal{C}_{\cdot \rightarrow 3}) = \text{cost} (\mathcal{C}^\star_3)$
> > > - $\text{cost} (\mathcal{C}_{\cdot \rightarrow 4}) = \text{cost} (\mathcal{C}^\star_4)$
> > > - $\text{cost} (\mathcal{C}_{\cdot \rightarrow 5}) = \text{cost} (\mathcal{C}^\star_5) + m$
> > >   - where $m =  \text{cost}(\mathcal{C}_{\cdot \rightarrow 3})$
> > > - $\text{cost} (\mathcal{C}_{\cdot \rightarrow 6}) = \text{cost} (\mathcal{C}^\star_6)$
> > >
> > > Once these precomputations performed (in $O(n \log n)$ for the Manhattan cost), one can proceed with Algorithm 2.
> > > The list of candidates is initialized with the pairs of neighbors in $z$ coming from distinct distributions:
> > > ```python
> > > list_candidates = [
> > >   # (sample1, sample2, cost)
> > >   (nu_1, mu_3, 0),
> > >   (mu_2, nu_1, 1),
> > >   (mu_3, nu_2, 1)
> > > ] # sorted in increasing cost
> > > ```
> > >
> > > Then, the loop over $k$ can start:
> > > - At iteration $k=1$,
> > >   - the active set becomes $\mathcal{A}_1 = \{ \nu_1, \mu_3\}$
> > >   - the cost for the solution at that stage is $\text{cost} (\mathcal{A}_1) = 0$
> > >   - there are two candidates left in `list_candidates` but they both overlap with $\mathcal{A}_1$, hence the list of candidates is emptied (line 7 of Algorithm 2) and a new candidate is added to the list (lines 8--11), with its marginal cost attached: `list_candidates = [(mu_2, nu_2, 2)]`
> > > - At iteration $k=2$,
> > >   - the active set becomes $\mathcal{A}_2 = \{\nu_1, \mu_3, \mu_2, \nu_2\}$
> > >   - the cost of the solution at that stage is $\text{cost} (\mathcal{A}_2) = \text{cost}(\mathcal{A}_1) + 2 = 2$
> > >   - A new candidate is then added to the list, with its marginal cost attached: `list_candidates = [(mu_1, nu_3, 4)]`
> > > - At iteration $k=3$,
> > >   - the active set becomes $\mathcal{A}_3 = \{\nu_1, \mu_3, \mu_2, \nu_2, \mu_1, \nu_3\}$
> > >   - the cost of the solution at that stage is $\text{cost} (\mathcal{A}_3) = \text{cost} (\mathcal{A}_2) + 4 = 6$
> > >
> > > As seen here, a single element from the candidate list is inspected at each iteration and the computation of the increase in cost relies on the marginal cost associated to this candidate and thus can be performed in $O(1)$ at each step.
> > >
> > > ### Discussion on transportation plans
> > >
> > > You are perfectly right that these iterations do not provide the *transportation plans* but rather the active sets for each iteration of the algorithm.
> > > Recovering the transportation plan for a given $k$ requires an additional sorting of the corresponding active set, which runs in $O(k \log k)$.
> > >
> > > Note however that our method provides active sets and optimal transport costs for all $k$ in $O(n \log n)$, as stated in the paper.
> > > In practice, if one needs the transportation plan at fixed $k$, the asymptotic complexity is unchanged.
> > > Moreover, given that the costs are available, the elbow method that we use in our experiments can also run in $O(n \log n)$ since the number of samples $k^\star$ to be transported is computed (using the elbow) based on costs only and the only transportation plan that needs to be built is the one corresponding to this $k^\star$, hence the overall process can be performed in $O(n \log n + k^\star \log k^\star) = O(n \log n)$ in this case too.

---

> > > > ### Author Response · Authors · 2024-12-02
> > > >
> > > > In terms of code, this would give:
> > > >
> > > > ```python
> > > > x = np.array([0., 1, 2])
> > > > y = np.array([-2., -1, 0])
> > > > indices_x, indices_y, marginal_costs = partial_ot_1d(x, y, max_iter=3, p=1)
> > > > k_star = 2  # Could be the elbow computed based on marginal costs
> > > > indices_x, indices_y = indices_x[:k_star], indices_y[:k_star]
> > > > marginal_costs = marginal_costs[:k_star]
> > > > print(indices_x, indices_y, marginal_costs)
> > > > #     [0 1]      [2 1]      [0.0, 2.0]
> > > > argsort_x = np.argsort(x[indices_x])
> > > > argsort_y = np.argsort(y[indices_y])
> > > > pi_k = np.zeros((3, 3))
> > > > pi_k[indices_x[argsort_x], indices_y[argsort_y]] = 1
> > > > print(pi_k)
> > > > # [[0. 1. 0.]
> > > > #  [0. 0. 1.]
> > > > #  [0. 0. 0.]]
> > > > ```
> > > >
> > > > Since we cannot update the paper anymore, if it is accepted, we propose adding a short paragraph to clarify and address any ambiguity regarding this important point you have raised.

---

### Official Review · Reviewer_aHWQ · 2024-11-02

**Soundness:** 3
**Presentation:** 3
**Contribution:** 3
**Rating:** 8
**Confidence:** 3

**Summary:**

The paper proposes a new algorithm to solve partial Wasserstein distance on the line which achieves the optimal complexity i.e., $\mathcal{O}(n\log n $ for $n$ is the number of supports. The key idea is to choose the best candidate set at each step of the algorithm. With the new algorithm to solve 1D partial Wasserstein, the paper proposes a new semi-metric named sliced partial Wasserstein which selects the projecting direction yielding the minimal transport cost under the lifted partial transportation plan. The paper shows that PAWL (the new algorithm) leads to faster computation than the previous unbalanced approach including optimal transport transport (OPT) and fast unbalanced optimal transport (Fast-UOT).  The paper conducts experiments in gradient flow and point cloud registration to show the efficiency of the proposed method.

**Strengths:**

* The paper addresses an important question in the community which is having a partial Wasserstein variant with the complexity of $\mathcal{O}(n\log n)$. Therefore, the proposed variant  can scale to more than 10000 supports.

* The proposed solution is novel including constructing and maintaining a candidacy set.

*  PAWL shows a favorable performance in computation, gradient flows, and point cloud registration compared to existing unbalanced optimal transport solver.

**Weaknesses:**

* The proposed algorithm requires the supports of two measures having the same weight. This setup might limit the application of the proposed algorithm i.e., the proposed algorithm seems to work well only for point clouds.

* The sliced partial Wasserstein is defined based on SWGG, however, it is still possible to define many other variants of sliced partial Wasserstein by using more projections

* There is no experiment for testing the robustness of the proposed methods and other baselines.

* The application seems to be limited to low-dimension i.e., <= 3.

**Questions:**

* Could elbow variant provide the true contamination percentage when the data contains additive noise?

* How could we extend the proposed approach to continuous cases and cases where measures do not having the same weight for their supports?

---

> ### Author Response · Authors · 2024-11-20
>
> ### Response to the following comment and question:
> > - The proposed algorithm requires the supports of two measures having the same weight. This setup might limit the application of the proposed algorithm i.e., the proposed algorithm seems to work well only for point clouds.
> > - How could we extend the proposed approach to continuous cases and cases where measures do not having the same weight for their supports?
>
> We thank the reviewer for raising these important questions regarding the broader applicability of our method. We address the two points separately below:
>
>   - **Extension to the Continuous Case**:
>
>     Extending PAWL to continuous measures is indeed an interesting and challenging problem. As our current method operates explicitly on discrete samples and relies on the pairwise computation of marginal costs, its direct applicability to continuous cases is non-trivial. We conjecture that it would require a different view of our iterative scheme, that could rely on the dual formulation of partial OT and the use of the Lagrangian formulation, akin to [6]. We view it as a promising direction for future research.
>
>   - **Extension to Unequal Weights**:
>     One should first notice that, in this paper, we focus on equal weights as it is the assumption that is made in most of the ML applications.
>     Then, relaxing this assumption to accommodate discrete distributions with arbitrary weights (e.g., weighted samples) would be a meaningful extension, potentially broadening the applicability of the method.
>     While such an extension has not been implemented in the current work, we believe it should be feasible to accommodate it within the PAWL framework and could be integrated in future developments. In more details, we conjecture that a sample would leave the complement of the active set (denoted $\bar{\mathcal{A}_k}$ in the paper) not as soon as some of its mass has been transported but rather when its mass has been _entirely_ transported.
>     Moreover, the extraction of minimal chains becomes non trivial and the impact on the computational properties should be carefully studied. As such, we leave it for future works.
>
>   We have updated the discussion of these points in the conclusion (cf. Section 6). We thank the reviewer for highlighting these limitations and their potential impact on real-world applications.

---

> > ### Author Response · Authors · 2024-11-20
> >
> > ---
> > ### Response to the following comments:
> > > - The sliced partial Wasserstein is defined based on SWGG, however, it is still possible to define many other variants of sliced partial Wasserstein by using more projections
> > > - There is no experiment for testing the robustness of the proposed methods and other baselines.
> >
> > Thank you for pointing out the possibility of defining alternative formulations for sliced partial Wasserstein distances using different projection schemes. While we acknowledge that multiple definitions are possible, we argue that SWGG is particularly well-suited to the partial setting for the following reasons:
> >
> >   - Sparsity in Partial Optimal Transport Maps:
> >     SWGG ensures sparsity in the resulting transport maps, which is a desirable feature of partial optimal transport. This sparsity aligns with the partial nature of the problem, where only a subset of the mass is transported.
> >
> >   - Straightforward Estimation of Transport Maps:
> >     The SWGG framework directly provides an estimated transport map rather than a general transport plan. This property is particularly beneficial in applications such as label propagation, where having an explicit map simplifies downstream tasks.
> >
> >   - Straightforward Maps for all masses $s$: by taking the minimum over different sampled lines, solutions for all masses $s$ can be obtained: it suffices to choose, for each amount of mass $s$, the line that provides the minimum OT cost.
> >
> >   - Experimental Validation of SWGG’s Advantages: To further demonstrate the utility of SWGG, we have included a new experiment in the revised manuscript that complements the gradient flow experiment. In this experiment, we compare SWGG to unbalanced optimal transport (UOT) with $L_2$ regularization in the context of outlier detection. Specifically, Figure 7 in the updated paper shows that UOT-based approaches often require setting a threshold on the transport plan to decide if a sample has been effectively transported. This thresholding can lead to inaccuracies, as evidenced by the fact that accuracies for UOT $L_2$​ do not converge to 0.5 (the baseline for a balanced classification). In contrast, the SWGG-based approach inherently provides a transport map, enabling exact label propagation without the need for thresholding, which leads to superior performance in this task.
> >
> >   - Robustness of our Sliced-PW Scheme: projection-based OT variants, such as Sliced-Wasserstein, is known to be more robust to data contamination than Vanilla Wasserstein, especially when considering $L_1$ norm (a thorough discussion is provided in [7]). Our Sliced-PW places itself in this context; nevertheless, we leave the theoretical analysis of this property for future work.
> >
> >
> >   We believe these points highlight the advantages of SWGG for the partial optimal transport problem and justify its use in our proposed framework. We thank the reviewer for their insightful comment, which has allowed us to strengthen the discussion in the revised manuscript.
> >
> > ---
> > ### Response to the following comment and question:
> > > - The application seems to be limited to low-dimension i.e., <= 3.
> > > - Could elbow variant provide the true contamination percentage when the data contains additive noise?
> >
> > It is at the moment unclear to us how we could theoretically study the ability of the elbow method to capture the contamination ratio in the presence of additive noise. Note that Partial or Unbalanced OT do not perform outlier detection: they remove samples that have large distance to the samples of the other distribution. Hence, the effectiveness of the elbow method probably depends on "how far" the outliers are from the original distribution. To our knowledge, this is still an open question in the optimal transport community.
> >
> >   We have included a new experiment that focuses on Domain Adaptation in the presence of outliers that further empirically showcases the benefits of computing solutions to all partial problems at once in practice in a high dimensional context.
> >
> > ---
> > [6] Luis A Caffarelli and Robert J McCann. Free boundaries in optimal transport and Monge-Ampere obstacle problems. Annals of mathematics, pp. 673–730, 2010.
> >
> > [7] Nietert, S., Goldfeld, Z., Sadhu, R., & Kato, K. (2022). Statistical, robustness, and computational guarantees for sliced wasserstein distances. Advances in Neural Information Processing Systems, 35, 28179-28193.

---

> > > ### Comment · Reviewer_aHWQ · 2024-11-23
> > >
> > > I would like to thank the authors for a detailed response on my questions. I raised my score since I believe the paper solved a special case of an important problem. I suggest the authors to include the discussion on other ways to define the sliced distances for completeness e.g., [1], [2], [3].
> > >
> > > [1] Max-Sliced Wasserstein Distance and its use for GANs, Deshpande et al.
> > >
> > > [2]   Energy-Based Sliced Wasserstein Distance, Nguyen et al.
> > >
> > > [3] Quasi-Monte Carlo for 3D Sliced Wasserstein, Nguyen et al.

---

> > > > ### Author Response · Authors · 2024-11-24
> > > >
> > > > We would first like to thank the reviewer for their feedback on our response.
> > > > We have included a short discussion about these approaches in an updated version of our paper (Section 4):
> > > >
> > > > > Other variants of Sliced Wasserstein have been defined that either focus on refining the averaging process (Nguyen & Ho, 2024; Nguyen et al., 2024) or build upon the maximum distance over sampled projection directions (Deshpande et al., 2019).

---

### Official Review · Reviewer_yYXF · 2024-11-04

**Soundness:** 3
**Presentation:** 2
**Contribution:** 2
**Rating:** 6
**Confidence:** 3

**Summary:**

This paper introduces PAWL (PArtial Wasserstein on the Line), an efficient algorithm for computing partial Wasserstein distances in one-dimensional distributions. The primary motivation is to improve the performance of optimal transport (OT) in scenarios where input distributions contain noise, outliers, or mass mismatches. PAWL enables exact partial transportation, computing solutions for all possible transported mass amounts simultaneously with an optimal time complexity of O(nlogn), making it suitable for large-scale applications.
Experiments confirm PAWL's computational efficiency and demonstrate its advantages in gradient flows and point cloud registration tasks.

**Strengths:**

* Paper is very-well written and clear
* Paper proposes a novel algorithm for efficiently calculating the partial 1-D Wasserstein on the lines in $O(nlogn)$

**Weaknesses:**

* Experimental designs lacks high-dimensional benchmarks and and PAWL is only tested on toy distributions.
* Calculating the chain costs is limited to $L_1$ norm and is unable to generalize to $L_2$ norms.
* PAWL’s incremental pair selection could face challenges when multiple candidates have nearly identical costs, which may lead to ambiguities in choosing the most optimal pairs.

**Questions:**

* I would like to investigate the effectiveness of the proposed method in higher-dimensional experiments and settings.
* How does PAWL ensure efficiency when managing the candidate list of pairs in each step, especially as the list grows with larger datasets?
* What mechanisms are in place to handle cases where multiple candidate pairs have nearly identical marginal costs?
When several pairs have similar transport costs, selecting the best pair could become computationally challenging or introduce instability

---

### Official Review · Reviewer_U54C · 2024-11-04

**Soundness:** 3
**Presentation:** 3
**Contribution:** 3
**Rating:** 6
**Confidence:** 3

**Summary:**

The paper discusses a novel algorithm and a slicing scheme to efficiently compute PArtial Wasserstein distances on a Line (PAWL) under L1 (Manhattan) cost  which is a variant of the linear assignment problem in $O(nlogn)$. The concept of "chains" that are specific patterns formed by the samples on the real line is introduced which enable fast computation of partial Wasserstein distances requiring a sorting operation as precomputation enabling fast computation of marginal and cumulative costs (sums). Authors develop a sliced partial Wasserstein (sliced-PW) scheme to enable sparse approx transport maps thus allowing identification of outliers (OOD samples) without requiring prior knowledge of the amount of noise/outliers. The proposed algorithm is then used in two applications - Wasserstein gradient flow and point cloud registration

**Strengths:**

- Extends several prior theoretical results to meaningfully solve partial 1-Wasserstein problem on line
- Introduces concept of chains and efficient way to calcualte marginal/cumualitve sums of cost for chains using precomputing step in O(nlogn)
- Designed scheme sliced-PW for using PAWL in real world applications like gradient flow and point cloud registration

**Weaknesses:**

- More thorough experimental evaluation would be great
- Applications to PU learning or outlier detection might be a good fit
- Might not be friendly to backpropagation algorithm limiting use in context of deep learning / neural nets

**Questions:**

- Can you suggest some other applications and extensions for PAWL?
- Is the algorithm usable in training neural net architectures? What is the notion of gradient in this case?

---

> ### Author Response · Authors · 2024-11-20
>
> ### Response to the following comments and question:
> > - More thorough experimental evaluation would be great
> >  - Applications to PU learning or outlier detection might be a good fit
> >  - Can you suggest some other applications and extensions for PAWL?
>
> Three of the reviewers agree that the paper would benefit from an additional real-world experiment with data lying in higher dimensional space. We agree that this is an important improvement for the paper. To address these points, we extended our experiments by applying the proposed Sliced-PW method to unbalanced domain adaptation, a relevant task closely related to outlier detection, in which having the set of solutions for all masses is compulsory.
>
> Specifically, we adapted the setup from [1] to evaluate Sliced-PW's performance in detecting contaminated data through regularization. The experiment involved source data comprising MNIST digits (classes 0, 1, 2, 3, 200 points per class) and target data consisting of MNIST digits (classes 0, 1) mixed with Fashion MNIST digits (classes 8, 9). Using label propagation, we classified the target data for varying mass amounts $s$.
>
>  Our results, summarized in Figure 7 of the revised manuscript, demonstrate that:
>   - Sliced-PW is more than 200 times faster than UOT and 14 times faster than [2] which are the main methods that provide the entire set of solutions.
>   - Sliced-PW achieves comparable performance to both competitors or even superior outlier detection performance than UOT by avoiding threshold parameter tuning.
>
>  This experiment highlights the adaptability of Sliced-PW to real-world tasks and its utility in scenarios like outlier detection. The revised version of the paper includes this discussion in a new Sec. 5.4, the corresponding figure, and the code is included in the anonymized repository for the sake of reproducibility.

---

> > ### Author Response · Authors · 2024-11-20
> >
> > ----
> > ### Response to the following comment and question:
> > > - Might not be friendly to backpropagation algorithm limiting use in context of deep learning / neural nets
> > > - Is the algorithm usable in training neural net architectures? What is the notion of gradient in this case?
> > We thank the reviewer for raising this important concern regarding the compatibility of PAWL with gradient-based optimization and its use in neural network training.
> >
> > As noted in the literature (e.g., [3]), back-propagating through sorting-based algorithms like PAWL poses challenges due to the non-smooth nature of the operations involved. Specifically, if PAWL is treated as a function of some parameter $\theta$, its gradient can be expressed as:
> >
> >   $$ \nabla_\theta \text{PAWL}= \langle \pi^\star , \frac{\partial C}{\partial \theta} \rangle $$
> >
> >  where $\pi^\star$ is the optimal transport plan and $C$ is the cost matrix. However, this gradient is often unstable, making it unsuitable for direct use in standard gradient-based optimization methods.
> >   To address this, we can for instance leverage perturbed optimizers (e.g., [4]), which introduce noise to smooth the optimization landscape.
> >
> >  For illustration, consider a 2D setup where we compute the PAWL distance between two distributions $\mu$ and $\nu$, projected onto 1D as a function of the orientation $\theta$ of the projection direction. Using a perturbed optimizer, the smoothed objective $\text{PAWL}_\epsilon (\theta)$ is defined as:
> >
> > $\text{PAWL}_\epsilon (\theta) = \mathbb{E} _{z \sim N _{0, 1}} [\text{PAWL} (\theta + \epsilon z)] $
> >
> > where $\epsilon$ controls the magnitude of the perturbation.
> >   One can then use Stein's lemma to get the following expression for its gradient (cf. [5], Section 14.4.5):
> >
> > $\nabla _{\theta} \text{PAWL} _\epsilon = \mathbb{E} _{z \sim N _{0, 1}} [(\text{PAWL} (\theta + \epsilon z) - \text{PAWL} (\theta)) \cdot z / \epsilon ]$
> >
> > The figure 9 in Appendix illustrates this approach in a simple 2D example, showing the smoothed PAWL in function of the orientation $\theta$ (top) and its gradient (bottom) with Monte Carlo estimation using 1k samples and $\epsilon=0.1$. We can see that this perturbation is sufficient to smooth the gradients, making them more stable and suitable for gradient-based optimization in neural network contexts.
> >
> >   While this approach demonstrates feasibility, further exploration is warranted to optimize stability and scalability in high-dimensional neural network settings. We appreciate the reviewer's feedback and believe this opens interesting avenues for future work.
> >
> >   This discussion is included in Appendix A.3 of the updated paper.
> >
> > ----
> >
> > [1] Laetitia Chapel, Rémi Flamary, Haoran Wu, Cédric Févotte, and Gilles Gasso. Unbalanced optimal transport through non-negative penalized linear regression. Advances in Neural Information Processing Systems, 34:23270–23282, 2021.
> >
> > [2] Abhijeet Phatak, Sharath Raghvendra, Chittaranjan Tripathy, and Kaiyi Zhang. Computing all optimal partial transports. In International Conference on Learning Representations, 2023.
> >
> > [3] Mathieu Blondel, Olivier Teboul, Quentin Berthet, and Josip Djolonga. Fast differentiable sorting and ranking. In International Conference on Machine Learning, pp. 950–959. PMLR, 2020.
> >
> >
> > [4] Quentin Berthet, Mathieu Blondel, Olivier Teboul, Marco Cuturi, Jean-Philippe Vert, and Francis Bach. Learning with differentiable pertubed optimizers. Advances in neural information processing systems, 33:9508–9519, 2020.
> >
> > [5] Mathieu Blondel and Vincent Roulet. The elements of differentiable programming. arXiv preprint arXiv:2403.14606, 2024.

---

> ### Author Response · Authors · 2024-11-26
>
> Following our response, we want to ensure that the clarifications we provided adequately address your questions and comments. If any aspect remains unclear or requires additional explanation, please do not hesitate to let us know. We would be happy to provide further details or rephrase certain parts to ensure that all your concerns are fully addressed. In particular, your question regarding the ability of PAWL to be used in neural net architectures was highly relevant, and we would like to ensure that our response has fully addressed any doubts.
> Once again, we sincerely appreciate your valuable contribution to improving our manuscript.

---

### Author Response · Authors · 2024-11-20
**Common response to all reviewers**

We would first like to thank the reviewers for their valuable inputs. Their insightful comments have led to the following revisions in the paper:

- As suggested by reviewers U54C, yYXF, and aHWQ, we have added a new experiment using high-dimensional data (images) to demonstrate the advantages of our Sliced-PW approach in an unbalanced domain adaptation context. This experiment is detailed in the newly added Section 5.4.

- In response to comments from reviewers yYXF and ErWA, we have clarified the complexity analysis of our PAWL solver for squared Euclidean cost (or any cost other than the Manhattan distance). We provide experimental validation of this complexity in Section 5.1. Additionally, a formal proof for the claimed complexities is now included in Appendix A.2.

- To address a question raised by reviewer U54C, we have added a discussion in Appendix A.3 regarding the differentiability of our PAWL solver.

All changes made to the paper are highlighted in purple in the updated version of the manuscript.

Below, you will find more specific responses to the individual comments of each reviewer.

---

> ### Comment · Reviewer_yYXF · 2024-11-20
>
> I would like to thank authors for addressing my concerns, and the new changes that they made for the paper.
>
> * At section 5.4, I understand that authors had limited time for the rebuttal, but MNIST and Fashion MNIST are not really considered as high-dimensional and complex tasks.
>
> * At section 5.1, although PAWL seems to be the fastet method, but for the common machine learning tasks where we generally use the $L_2$ norm, if the cardinality of the measure is small, OPT runs faster and in large cardinalities, Fast-UOT is still on par, though I understand that PAWL may outperform both of them.
>
> Unfortunately, I'm still not convinced about the scalability of the PAWL.

---

> > ### Author Response · Authors · 2024-11-21
> >
> > We thank the reviewer for their feedback and for taking the time to engage with our work. Below, we address the key points raised:
> >
> > - **On the scope of experiments**:
> >   While we acknowledge the comment regarding limiting time to provide new experiments, our submission already includes a comprehensive set: gradient flows, point cloud registration, and a new experiment on detecting data from different domains. The added experiment specifically aligns with the setup in [1] that has been used to validate the advantage of computing the entire set of solutions (for all values of the hyper-parameter) in a similar context. This experiment uses data of dimension 784, demonstrating the applicability of our approach in high-dimensional settings.
> >
> > - **Regarding timings and $L_1$ norm**:
> >   We respectfully disagree with the reviewer’s concerns about the $L_1$ norm. Its relevance in machine learning is well-established (e.g., WGAN) and it is often stated to be more robust (e.g. see [7] in the context of sliced Wasserstein).
> >   PAWL is particularly compelling because:
> >     - It provides all solutions across parameter values, which eliminates the need for repeated runs that may be required by competitors (e.g., during cross-validation).
> >     - Experiments demonstrate that PAWL (using the $L_1$ norm) is more than one order of magnitude faster than competitors. This efficiency remains significant even when extended to the $L_2$ norm.
> >     - Additionally, we emphasize that timing results are inherently implementation-dependent. As a consequence, the relative slope of the runtime curves across methods is more indicative of the scalability and theoretical efficiency of PAWL than the absolute values, which may vary due to implementation details.
> >       Looking at the slopes, PAWL $W_1$ is competitive with Fast-UOT and PAWL $W_2^2$ outperforms OPT.
> >
> > Finally, if the reviewer thinks that it is useful, we can add a discussion about the choice of $L_1$ as a ground cost in the core of the paper, as well as a further analysis of the runtime.
> >
> > ---
> > [1] Laetitia Chapel, Rémi Flamary, Haoran Wu, Cédric Févotte, and Gilles Gasso. Unbalanced optimal transport through non-negative penalized linear regression. Advances in Neural Information Processing Systems, 34:23270–23282, 2021.
> >
> > [7] Nietert, S., Goldfeld, Z., Sadhu, R., & Kato, K. (2022). Statistical, robustness, and computational guarantees for sliced wasserstein distances. Advances in Neural Information Processing Systems, 35, 28179-28193.

---

> > > ### Comment · Reviewer_yYXF · 2024-11-25
> > >
> > > I understand taht $L_1$ can be considered as a special case of this interesting problem. Although I expected to see a better experimental setting in this paper, authors address an interesting problem and I'm fairly satisfied with their rebuttal. As a result, I raised my score to 6.

---

> > > > ### Author Response · Authors · 2024-11-26
> > > >
> > > > Thanks for your response, we appreciate your endorsement.

---

### Meta-Review · Area_Chair_wGi1 · 2024-12-18

**Metareview:**

This paper presents PAWL, an efficient algorithm for computing partial Wasserstein distances on the line, achieving a time complexity of O(n log(⁡n)). The primary contributions include the flexibility of solving the problem for all admissible transported mass amounts simultaneously and a novel slicing strategy tailored to avoid transporting mass between noisy data points. Reviewers acknowledged the algorithm's computational efficiency and its application to tasks like gradient flows and point cloud registration. However, several weaknesses were identified that should be addressed to strengthen the final version. One notable concern, highlighted by ErWA, is the lack of a formal complexity statement for the algorithm as proposed in Proposition 7, which currently lacks clarity. A precise statement, such as confirming O(n^2) complexity when the cost is a metric, was suggested alongside proof. Other concerns include limited experiments on high-dimensional data, insufficient robustness testing, and constraints in adapting the method to non-metric costs or continuous settings. The rebuttal addressed some of these issues, such as providing additional complexity clarifications and adding experiments with higher-dimensional data, though the data remained relatively simple (e.g., MNIST). The final version should incorporate these discussions comprehensively, with rigorous theoretical and experimental validation. Based on the authors' responsiveness and the algorithm's potential impact, this paper merits acceptance.

**Additional Comments On Reviewer Discussion:**

During the rebuttal period, reviewers raised concerns about the clarity of the algorithm's complexity, the limited scope of high-dimensional experiments, and the robustness of the proposed method. The authors responded by clarifying the complexity analysis, including a formal proof in the appendix, and adding new experiments on higher-dimensional datasets, although these datasets were relatively simple (e.g., MNIST). While the updates addressed some concerns, questions about scalability and robustness testing persisted. These discussions were carefully weighed, and the decision to recommend acceptance was based on the significant computational contribution, the authors' responsiveness, and the potential impact of the proposed method, with the expectation that the final version will fully address the remaining issues.

---

### Decision · Program_Chairs · 2025-01-22

Accept (Poster)